# Deep immune profiling delineates hallmarks of disease heterogeneity in extrapulmonary tuberculosis

Sebastian J. Theobald[1,2,3,12], Kilian Dahm [4,5,6,12], Dinah Lange [1,2,3,12], Jannis B. Spintge [4], Sandra Winter[1,2], Angela Klingmüller[1,2,3], Lisa Holsten [4,5], Alexander Simonis [1,2,3], Elena De Domenico [4,7], Henning Walczak [8,9,10], Martina van Uelft[4,5], Joachim L. Schultze [4,5,7], Marc D. Beyer [4,7,11], Thomas Ulas[4,5,7,12] ✉, Isabelle Suárez[1,2,3,12] & Jan Rybniker [1,2,3,12] ✉

Our understanding of the immune response in tuberculosis (TB) remains incomplete. This applies in particular to extrapulmonary TB (EPTB), a highly heterogeneous disease affecting up to 30% of patients in certain regions. Based on data-driven clustering of blood transcriptomes in an EPTB patient cohort, we define three highly distinct immunotypes. Combining bulk with single-cell RNA-sequencing delineates immunological trajectories characterized by dynamic IFN- and IL-1-mediated signalling in monocytes, alongside hyperactivation of T and NK cells, ultimately resulting in extensive immune dysregulation. Integrative analysis of multi-omics data provides deep insights into different layers of the anti-tuberculous immune response and the identification of immunotypes enabling stratification strategies for personalized host-directed treatments. In addition, our comprehensive approach helps to develop an accurate diagnostic gene expression signature for both EPTB and pulmonary TB highlighting the translational potential of our data.

Mycobacterium tuberculosis (*Mtb*), the causative agent of tuberculosis (TB), remains a significant contributor to both mortality and morbidity worldwide. In 2023 alone, *Mtb* caused 1.25 million deaths and more than 10 million new infections, despite extensive global efforts to control the disease[1].

Improving our understanding of disease pathogenesis and how *Mtb* modulates human immune responses is crucial to enhance our diagnostic, prophylactic, and therapeutic efforts against TB. In the past decade, research in this field has focussed mainly on pulmonary TB (PTB), a contagious form of the disease affecting the lungs.

In contrast, extrapulmonary TB (EPTB), which accounts for 16% of global TB cases and up to 30% of cases in several countries, has been scientifically understudied[1]. EPTB can involve various organ systems other than the lungs, including the lymph nodes (50% of all EPTB cases)[2],

[1]Department I of Internal Medicine, Medical Faculty and University Hospital Cologne, University of Cologne, Cologne, Germany. [2]Center for Molecular Medicine Cologne (CMMC), Medical Faculty and University Hospital Cologne, University of Cologne, Cologne, Germany. [3]German Center for Infection Research (DZIF), Bonn-Cologne, Germany. [4]Systems Medicine, German Center for Neurodegenerative Diseases (DZNE), Bonn, Germany. [5]Genomics and Immunoregulation, Life & Medical Sciences (LIMES) Institute, University of Bonn, Bonn, Germany. [6]Translational Pediatrics, Department of Pediatrics, University Hospital Wuerzburg, Würzburg, Bavaria, Germany. [7]Platform for Single Cell Genomics and Epigenomics at the German Center for Neurodegenerative Diseases, the University of Bonn and West German Genome Center (WGGC), Bonn, Germany. [8]Centre for Cell Death, Cancer, and Inflammation (CCCI), UCL Cancer Institute, University College London, London, UK. [9]Institute of Biochemistry I, Medical Faculty, University of Cologne, Cologne, Germany. [10]CECAD Research Centre, University of Cologne, Cologne, Germany. [11]Immunogenomics & Neurodegeneration, German Center for Neurodegenerative Diseases (DZNE), Bonn, Germany. [12]These authors contributed equally: Sebastian J. Theobald, Kilian Dahm, Dinah Lange, Thomas Ulas, Isabelle Suárez, Jan Rybniker. ✉e-mail: t.ulas@uni-bonn.de; jan.rybniker@uk-koeln.de

bones, joints, central nervous system, genitourinary tract, gastrointestinal system, or skin. In the most severe form of the disease, *Mtb* disseminates to multiple organs, which often causes critical illness[3,4].

Due to the heterogeneous clinical presentation, EPTB can be challenging to diagnose, and diagnostic procedures often require invasive techniques such as tissue biopsies to detect *Mtb*[3]. This stands in contrast to contagious forms of PTB, where confirmation of the disease and monitoring of a successful treatment response can be performed by serial microbiological examination of sputum smears. Such easy-to-obtain bio-samples are not available in EPTB.

In the past decade, research on blood transcriptional and immune profiling has strongly improved our understanding of host-pathogen interactions in infectious diseases[5,6]. Several studies on PTB provided deep insight into the characteristics of the human immune response, including its development in newly infected patients and its resolution during treatment[7–15]. Interestingly, some studies found that transcriptional profiles in TB reflect the extend of disease in different individuals[15,16]. Blood transcriptional studies also gave the first evidence that type I interferon (IFN) mediated signaling is associated with active TB. In addition, it was found that the cytokines IFN-α/β are detrimental to a protective immune response, primarily by driving neutrophil-mediated inflammation[17,18]. Further, IFN signaling has been reported to be closely interconnected to interleukin 1 (IL-1) signaling and immunotherapeutic modulation of these pathways altered the susceptibility of mice to *Mtb*[17,19]. Nevertheless, it is not well understood how exactly type I IFNs and IL-1 are linked to disease severity and which cell types drive their expression in human disease. In parallel, it is known that T and NK cell responses are essential for the anti-tuberculosis immune response[18,20–23], however, the underlying immune programs of T and NK cells in relation to disease progression remain understudied. Addressing these gaps will be crucial for the successful implementation of host-directed therapies (HDT), which are needed to potentiate conventional antibiotic treatment or to abrogate hyper-inflammatory immune responses leading to long-term sequelae[24–26]. The effectiveness of such treatments has rarely been systematically investigated in EPTB. Given the heterogeneity of clinical presentations, it is plausible that stratification strategies based on immunological markers may be necessary to achieve positive outcomes in clinical trials assessing HDTs in EPTB.

Distinct transcriptional profiles found in the whole blood of active TB patients also supported the development of gene expression signatures for improved TB diagnosis. Reported blood signatures targeting variable numbers of differentially expressed genes have been tested for classifying active TB, latent TB, and other pulmonary diseases[6,8,9,13–15]. The first commercial triage test based on a three-gene transcriptomic signature has been developed and tested in pediatric and adult PTB patients[27,28]. Thus, blood-based transcriptomic and immunological signatures may serve as easily accessible biomarkers to diagnose EPTB, potentially having a significant impact on patient care.

In this study, we applied high-throughput whole blood RNA sequencing (RNA-seq) and high-resolution single-cell RNA-sequencing (scRNA-seq) combined with multi-color flow cytometry and cytokine multiplexing in a clinically well-defined and highly heterogeneous cohort of EPTB patients. Using an unbiased, multi-modal, and integrative approach, we were able to define EPTB patient groups with distinct immunotypes. Our comprehensive immunological characterization of EPTB patients (Fig. 1a) offers compelling insights into the underlying host-pathogen interactions of TB. The findings also hold promise for the development of highly accurate biomarkers and the advancement of personalized therapeutic approaches.

## Results

### Blood transcriptional profiling delineates distinct groups of EPTB patients

To investigate global gene expression shifts in the whole blood of patients diagnosed with EPTB, bulk RNA-seq was performed at the time of diagnosis and prior to initiation of antibiotic therapy. All participants in our study were confirmed to have active TB through microbiological and histological testing and were subjected to a standardized diagnostic protocol, which assessed inflammatory parameters, the involved organs and the overall severity of the disease as extensively described in the methods section (Supplementary Table 1)[29].

First, the transcriptomes of the entire EPTB cohort ($n = 29$) were compared to those obtained from a control group of healthy individuals ($n = 9$). Although some patients exhibited severe symptoms characterized by pronounced clinical inflammatory responses, only a small set of 84 up- and 3 downregulated genes were found to be differentially expressed in diseased individuals (Fig. 1b). This stands in noticeable contrast to recent studies on active PTB, which typically identify hundreds of differentially expressed genes (DEGs) compared to healthy controls[13,18]. Irrespective, the limited number of significantly upregulated genes in whole blood of all EPTB patients were linked to type I and II IFN-stimulated genes (ISGs) (*IFI6/44/27*) and inflammatory/complement signaling (*CD274* (transcript of PD-L1), *C1QA/B/C*) as well as genes coding for Fcγ-receptors (*FCGR1A* and *1B*) (Fig. 1c, d and Supplementary Fig. 1a).

Considering the heterogeneity of EPTB leading to distinct clinical phenotypes, we implemented a data-driven strategy for clustering of individual transcriptional profiles. First, we identified the 9013 most variable genes based on the inflection point of a gene expression variance ranking curve. Clustering this gene subset stratified the EPTB patients into three distinct groups (*group 1*: 13 patients; *group 2*: 11 patients; *group 3*: 5 patients) (Fig. 1e). Subsequently, we assessed transcriptional alterations across patients by performing a principal component analysis (PCA). A profound difference between healthy individuals and EPTB patients was identified, and the three EPTB groups were separated across PC1 (Fig. 1f). We matched clinical imaging data and inflammatory laboratory parameters such as C-reactive protein (CRP) and interleukin 1 (IL-1) with the transcriptomic data-driven patient clustering results. The majority of group I patients (9 out of 13, 69%) exhibited isolated cervical lymph node TB, a mild form of EPTB (Supplementary Table 1), while the majority of group II patients presented with TB in multiple lymph nodes (Supplementary Table 1), and all group III patients suffered from disseminated disease affecting various organs, bones, and lymph nodes. None of the severely ill patients had concurrent conditions, such as HIV/AIDS, offering alternative explanations for the severity of their illness. Interestingly, CRP values differed significantly between Group 1 and Groups 2 and 3 (Fig. 1g). In contrast, the highly pro-inflammatory cytokine IL-1β, which is a well-known driver of TB pathogenicity[30], was significantly upregulated in group 3 patients only (Fig. 1g). Other pro-inflammatory cytokines showed similar patterns with significant differentiation of the three groups which were separated based on blood transcriptomes (Fig. 1h and Supplementary Fig. 2).

Based on these transcriptomic and clinical parameters, we categorized the three immunotypes as "low inflammation (INF$_{low}$)", "intermediate inflammation (INF$_{int}$)", and "high inflammation (INF$_{high}$)". In INF$_{low}$ patients, we detected only 16 upregulated DEGs in comparison to healthy controls (Fig. 1i and Supplementary Fig. 1b). These DEGs included type I IFNs and *CD274* (transcript of PD-L1), which overlapped with the DEGs detected in our global approach including the entire cohort (Fig. 1c). A markedly higher number of DEGs was detected in INF$_{int}$ EPTB patients (group II) with 285 upregulated and 115 downregulated DEGs (Fig. 1j and Supplementary Fig. 1b). These were primarily linked to inflammatory signaling (e.g., *C1QA/B/C*, *CCL3*, *STAT1*, *CD14*, *GZMB*, *CASP1*), metabolic processes (*ACOD1*, *CMPK2*) and the type I IFN-driven signature observed in the INF$_{low}$ group (Fig. 1i). INF$_{high}$ EPTB patients displayed substantial transcriptional alterations, with 1238 upregulated and 821 downregulated DEGs (Fig. 1k and Supplementary Fig. 1b). Importantly, this stratification substantially

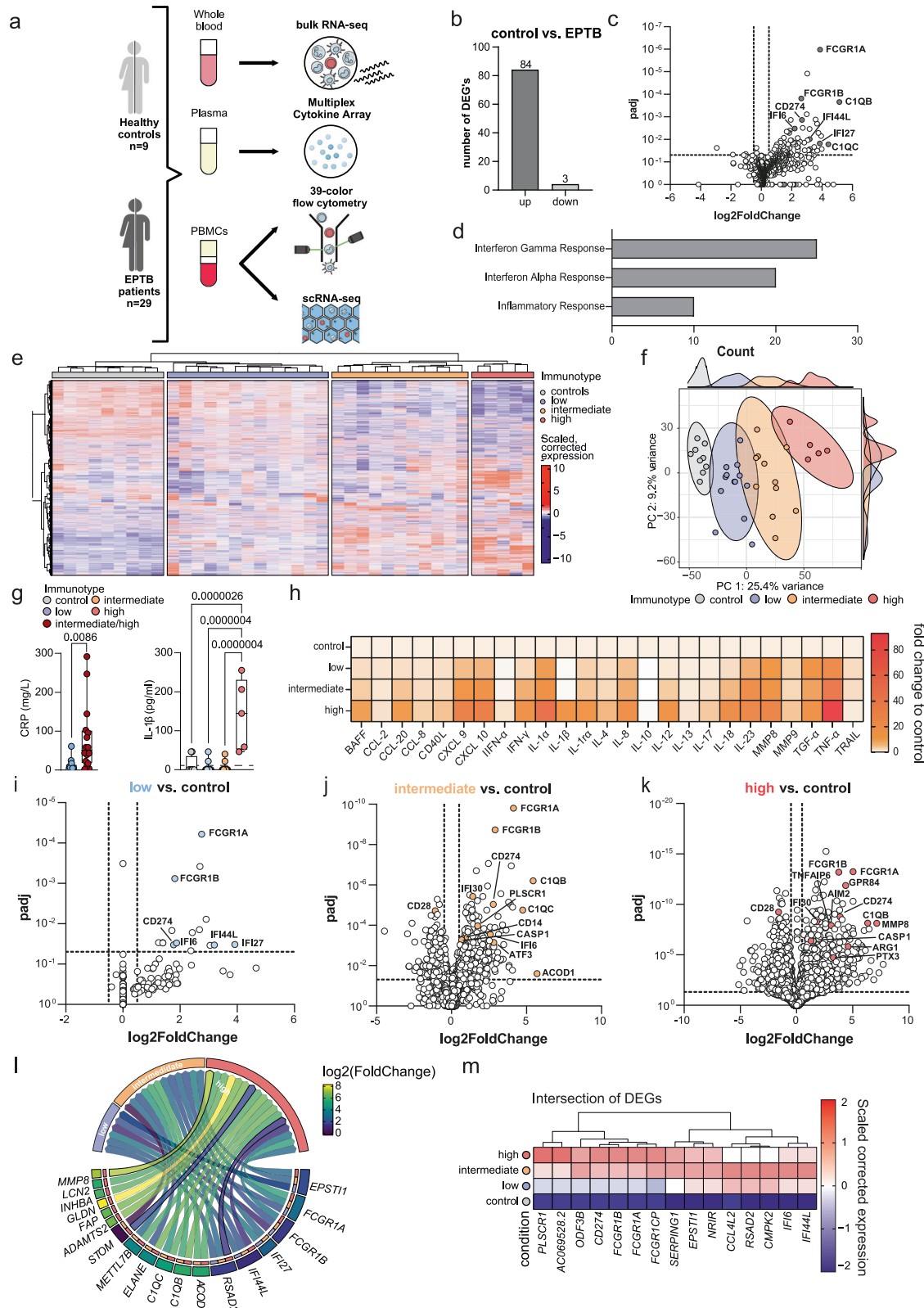

reduced within-group variance, thereby validating the utility of our clustering approach, as evidenced by a marked increase in the number of DEGs. Changes between the control group and INF$_{high}$ EPTB patients primarily involved an increased expression of pro-inflammatory signaling, inflammasome, and immune cell activation genes such as *MMP8, AIM2, CASP1, CD177*, and at the same time, a reduced expression of genes related to the adaptive immune response was observed

(*CD28, CD19, CD79A, IL7R*) (Fig. 1k). Leading-edge analysis from all EPTB groups revealed that IFN-associated genes such as IFI44L were differentially expressed in all immunotypes, indicating differential IFN signaling as a general hallmark of EPTB. In contrast, inflammatory genes such as *ELANE* and *MMP8* were expressed in INF$_{int}$ or INF$_{high}$ EPTB, while the intersection of the downregulated genes of INF$_{int}$ and INF$_{high}$ patients were involved in B cell activation (Fig. 1l and

**Fig. 1 | Whole-blood RNA-sequencing analysis for detection of EPTB immunotypes. a** Scheme of the study. Created in BioRender. Dahm, K. (2025) https://BioRender.com/iodspuu. **b** Number of all significant up- and downregulated differentially expressed genes (DEGs) (absolute (FC) ≥ 2, Benjamini-Hochberg-adjusted *p*-value < 0.05; applies to all following DEGs) in EPTB compared to healthy controls. **c** Volcano plot of DEGs in (**b**). **d** Gene set-enrichment analysis of DEGs from b) using the Molecular Signature Database (MSigDB) Hallmark terms. **e** Heatmap of the scaled, batch-corrected expression of the 9013 most variable genes (rows) based on the inflection point of a gene variance-ranking curve per patient (columns). Patients were grouped into $INF_{low}$ (*n* = 13), $INF_{int}$ (*n* = 11) and $INF_{high}$ (*n* = 5) EPTB immunotypes based on a hierarchical clustering analysis. **f** Principle component analysis (PCA) for most variable genes. **g** Boxplot of CRP and IL-1β measurements in plasma of EPTB patients. For the CRP measurement, the $INF_{int}$ and $INF_{high}$ patients were pooled. *P*-values were calculated with an unpaired t-test (CRP) or a one-way-ANOVA with Turkey´s post-test (IL-1β). Boxplots show the 25%, 50% (median) and 75% percentile, whiskers range up to the lowest/highest points. CRP: low *n* = 13 donors, intermediate/high *n* = 16 donors; IL1β: control *n* = 8 donors, low *n* = 11 donors, intermediate *n* = 11 donors, high *n* = 5 donors. **h** Heatmap (fold change to control) of plasma cytokine and chemokine bead-array based measurements of all EPTB patients and controls. Significances are measured with a two-way-ANOVA with Dunnett's post-test. **i–k** Volcano plot of all significant up- and downregulated DEGs in $INF_{low}$ EPTB, $INF_{int}$ EPTB and $INF_{high}$ EPTB. **l** Circos plot of the union of DEGs part of the leading edge of each comparison, defined as the genes above 75 % of the maximal log2 fold change or maximal inverted log10 transformed adjusted *p*-value. Arrows are colored according to the log2FC of the respective comparison, and segments display the mean log2FC per gene. **m** Heatmap of the intersection of DEGs of all comparisons colored according to the mean batch-corrected and scaled expression value. Source data are provided as a Source Data file for the respective figures.

Supplementary Fig. 1c). Interestingly, when quantifying overlapping DEGs, only 15 genes were shared between all EPTB groups (Fig. 1m). A functional enrichment analysis showing enrichment for anti-viral responses in $INF_{low}$ EPTB underlined the prominent type I IFN-associated gene signature observed in this patient group. In $INF_{int}$ and $INF_{high}$ EPTB, we found strong enrichment for type I/II IFN together with IL-1 signaling and other pro-inflammatory pathways (Supplementary Data 1).

## Global transcriptomic signatures reflect distinct changes in the gene expression profiles dissecting biological processes in EPTB patients

To gain deeper insights into the complex regulatory landscape in EPTB, we aimed to identify modules of co-expressed genes and the associated biological processes that further characterize our stratified EPTB patient groups. We employed the construction of co-expression networks and analysis (hCoCena)[31]. The final hCoCena network was comprised of 4,965 genes distributed across ten modules (color-coded), each characterized by unique gene profiles that were either specific to one or to multiple EPTB phenotypes (Fig. 2a). Six out of ten modules showed significant differences in their module gene expression across healthy controls and EPTB patients and were further characterized with functional enrichment analyses of the module genes based on Gene Ontology (GO) and Kyoto Encyclopedia of Genes and Genomes (KEGG) databases (Fig. 2b, c). The wheat module, primarily expressed in healthy individuals and $INF_{low}$ EPTB patients (Fig. 2a), showed enrichments associated with T and B cell activation and antigen-presentation (e.g., *CD3E, CD28, CCR7, CD40LG*) (Fig. 2c,), indicating a functional adaptive immune response. The gold module genes depicted elevated expression levels in $INF_{low}$ and to a greater extent in $INF_{int}$ EPTB patients (Fig. 2a). This module was characterized by the enrichment of genes involved in immune responses mediated by T and NK cells (*KLRK1, KLRD1, GZMB*), along with an inflammatory response (*CCL2/5, CCR5*; chemokine and cytokine signaling) (Fig. 2c, d). The turquoise and lightgreen modules portrayed immune responses of both $INF_{int}$ and $INF_{high}$ EPTB, while plum and coral modules were associated with $INF_{high}$ EPTB patients only. Plum and coral were associated with an antibacterial and neutrophil-driven immune response together with strong activation of pro-inflammatory signaling (TLR and JAK-STAT signaling) (Fig. 2c). Significant involvement of the cellular metabolism was represented in the lightgreen module (Fig. 2a, c). The turquoise module was characterized by IFN- (I and II), NOD-like, TNF- and IL-1-signaling, expressing a variety of immune cell type markers (*TLR2, TLR4, TLR6, NOD2*) (Fig. 2c, d). Thus, the turquoise together with the plum module indicates a hyper-inflammatory immune response involving inflammasome and pyroptosis (*IL1B, NLRP3*) associated signaling pathways observed almost exclusively in $INF_{high}$ EPTB patients (Fig. 2c, d). Interestingly, the enrichment for cytokine-cytokine receptor interaction was shared by five (gold, wheat, turquoise, seagreen, and plum) out of ten modules, suggesting that cytokine signaling is one of the major drivers in EPTB immune pathology (Fig. 2c). For delineation of compositional changes within major immune cell types, we deconvoluted our data utilizing published single-cell RNA-seq data[32] (Supplementary Fig. 3a). CD14+ and CD16+ monocytes significantly increased with immunotypes while an immature neutrophil fraction was observed almost exclusively in $INF_{high}$ EPTB (Fig. 2e and Supplementary Fig. 3b). In contrast, CD4+ T cells and B cells significantly decreased in more inflammatory stages, whereas CD8+ T cells and NK cells showed no differences (Fig. 2e). Taken together, we established an innovative bioinformatic pipeline capable of unraveling distinct immune responses and immune alterations linked to different inflammatory stages in EPTB patients.

## Multi-dimensional flow cytometry analyses delineate immunological programs linked to disease severity

To associate findings derived from whole blood transcriptomes with cytometry-based immune cell phenotypes, we performed multicolor flow cytometry (gating strategy shown in Supplementary Figs. 4 and 5) in conjunction with population clustering methods to identify immunotypes associated with distinct transcriptomes and immune compositions in each EPTB severity stage.

Overall, we found an expansion of myeloid cells and NK cells (CD3−CD19−) and a reduction of T and B cells with increasing disease severity (Fig. 3a and Supplementary Fig. 6a, b). Quantification of the innate immune cell compartments revealed that the fractions of activated monocytes (HLA-DR+), classical monocytes, and intermediate monocytes gradually increased, while NK cells and DCs declined (Supplementary Fig. 6c, d). Flow-self-organizing maps (SOM) analyses of monocytes revealed in total 10 clusters, of which 5 clusters were chosen for downstream analysis, as they represented relevant monocyte marker expressions and sufficiently high cell frequencies (Fig. 3b and Supplementary Fig. 6e–g). In line with the upregulation of monocyte-associated genes in our bulk RNA-seq analysis, activated monocytes (Cluster 7) showed a gradual expansion in the respective immunotypes, whereas activated non-classical monocytes (Cluster 8) and activated intermediate monocytes (Cluster 9) peaked in $INF_{int}$ EPTB (Fig. 3b and Supplementary Fig. 6h). The surface expression of myeloid activation markers CD80 and CD86 increased towards the $INF_{high}$ immunotype of CD14+CD163+HLA-DR+ monocytes (Cluster 7) (Fig. 3c). PD-L1, a top upregulated gene in EPTB transcriptomes, was strongly expressed on all analysed myeloid cell types independent of the grouping (Fig. 3d).

Corroborating the findings of differential NK cell functionality identified in the bulk RNA-seq data (gold module), we analysed the activating surface receptors NKG2D and NKp44 (Fig. 3e and Supplementary Fig. 6i). Using cluster analysis and MFI quantifications, we found that CD56dim and CD56hi NK cells significantly upregulated both receptors, peaking in $INF_{int}$ EPTB (Fig. 3e, f and Supplementary Fig. 6i)

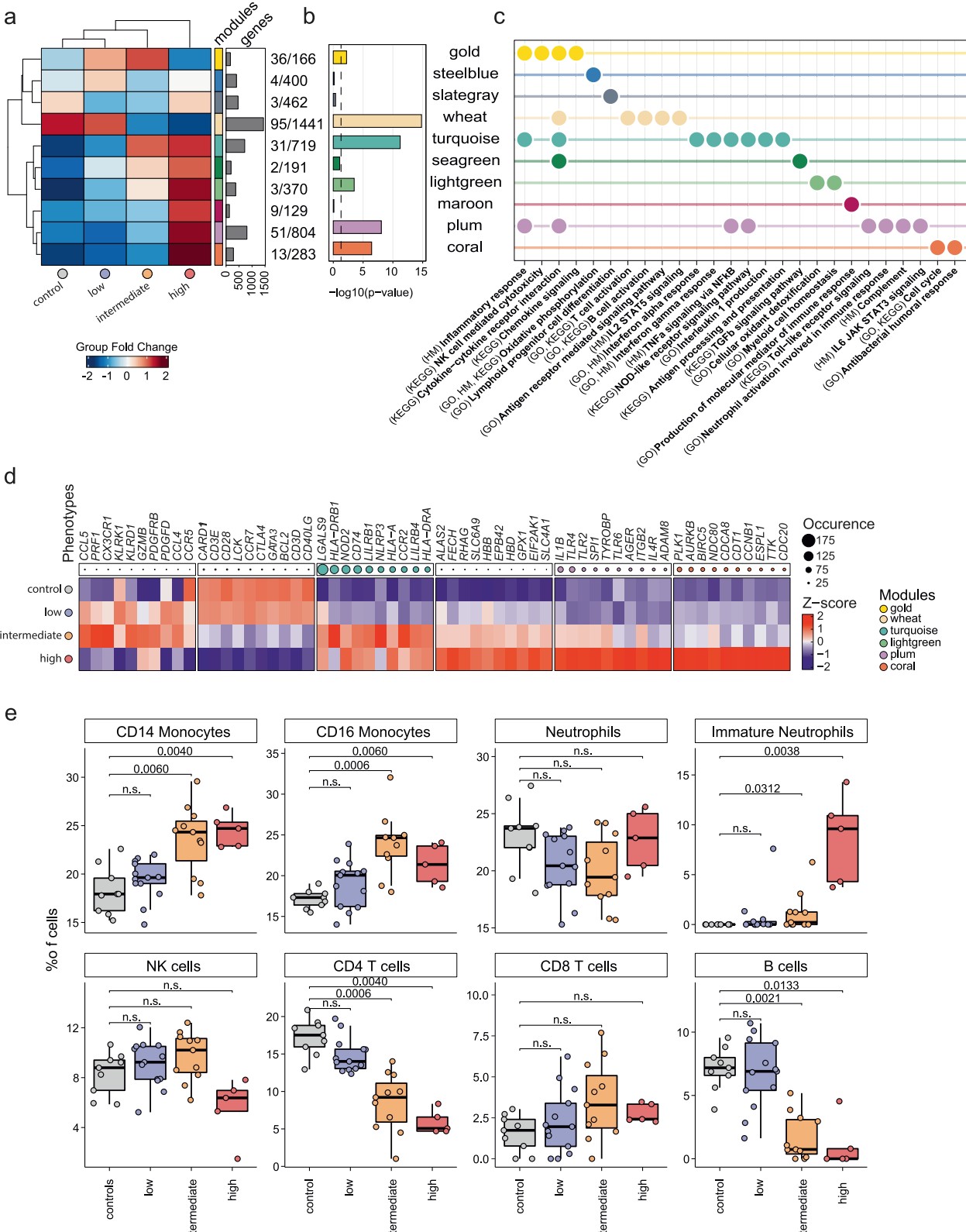

and confirming enhanced phenotypic and transcriptomic NK cell activation in INF$_{int}$ EPTB, which decreased in INF$_{high}$ EPTB.

When addressing the T cell compartment, we observed a phenotypic shift from naïve T cells to effector memory and terminal effector T cells, which peaked in INF$_{int}$ EPTB (Fig. 3g and Supplementary Fig. 7a). Cluster analysis (Fig. 3h and Supplementary Fig. 7b–d) confirmed these phenotypic and transcriptomic changes (Fig. 2, gold

module), as the frequency of the clusters containing naïve CD8$^+$ and CD4$^+$ T cells (Cluster 1 and 12) decreased in EPTB patients compared to controls, especially in the INF$_{int}$ group (Supplementary Fig. 7d). Expression of the T cell activation markers TIGIT and PD-1 was elevated in all EPTB groups (Fig. 3i, j and Supplementary Fig. 7e). In particular, TIGIT expression peaked in the INF$_{int}$ patient group, whereas PD-1 was strongly expressed in INF$_{high}$ EPTB (Fig. 3i, j). As expected, terminal

**Fig. 2 | hCoCena analysis for detection of biological functions in EPTB immunotypes. a** Heatmap of modularized gene co-expression network colored by the group fold change (GFC) (left) and bar plot of the number of genes included in each gene module. Rows and columns are clustered using the complete clustering algorithm with Euclidean distances. **b** Bar plot of *p*-values calculated with a one-way ANOVA across the GFC of each stratified EPTB immunotype per module. The black dashed line represents the significance threshold of 0.05. **c** Dot plot of representative GO, KEGG and MSigDB Hallmark terms significantly overrepresented per module (Benjamini-Hochberg-adjusted *p*-value < 0.05). **d** Heatmap of mean batch-corrected and scaled expression of module genes with the most occurrences among all significant GO, KEGG and MSigDB Hallmark terms. Columns are annotated with dots scaled by the occurrences of the gene. The top 10 genes from the significant module in B were displayed. **e** Boxplot of the percentage of cells computed by cell type deconvolution based on a whole blood reference of healthy individuals from Schulte-Schrepping et al.[32] per sample and selected cell type, and colored by EPTB immunotype. Boxplots show the 25%, 50% (median) and 75% percentile, whiskers denote 1.5 times the interquartile range. Statistics were computed by an unpaired two-sided Wilcoxon test followed by a Benjamini-Hochberg adjustment. Control *n* = 9 donors, low *n* = 13 donors, intermediate *n* = 11 donors, high *n* = 5 donors. Source data are provided as a Source Data file for the respective figures.

effector T cells (Cluster 11) exhibited the highest expression levels of both markers (Supplementary Fig. 7f, g). Together, these data indicate strong T cell activation in INF$_{int}$ EPTB and signs of T cell hyperactivation.

B cells are increasingly recognized as modulators of the immune response towards *Mtb* infection[33,34]. Comparing the distribution of the B cell subtypes, it became obvious that all clusters (Cluster 2-6) containing IgG$^+$ or IgM$^+$ B cells declined in the INF$_{high}$ group compared to healthy controls (Supplementary Fig. 8a–d). Cluster 1 was excluded, as only minor B cell marker expression was observed. This effect was most pronounced in CD27$^+$ memory B cell clusters (Cluster 5 and 6), suggesting a potentially impaired B cell response in INF$_{high}$ EPTB (Supplementary Fig. 8d).

In conclusion, our comprehensive immune cell analysis provided robust phenotypic validation of the three distinct immunotypes identified by data-driven clustering and network analysis of blood transcriptomes.

## Single-cell RNA sequencing reveals dynamic leucocyte trajectories with major phenotypic switches of monocyte subsets

To decipher molecular immunotypes of EPTB on a single cell level, a total of 61,537 peripheral blood mononuclear cells (PBMCs) derived from three donors of each EPTB immunotype and healthy controls were analysed using scRNA-seq (Fig. 4a). Sex distribution was taken into account when selecting the donors for scRNA-seq. Cells were annotated based on cluster marker genes, protein marker expression and commonly used cell markers (Fig. 4b, Supplementary Fig. 9a and Supplementary Data 2). Differential expression analysis between all EPTB patients and controls showed a strong increase in upregulated genes within monocytes, in particular CD14$^+$ monocytes, and in CD4$^+$ and CD8$^+$ T cells, whereas mostly downregulated genes were identified in B cells (Fig. 4c). Enrichment analysis revealed enrichment of IFN-related genes (type I and II IFN) across all cell types and a strong enrichment of responses to bacteria and immune effector processes within monocytes and CD8$^+$ T cells (Fig. 4d). Moreover, enrichment of the downregulated genes showed an involvement in TNFα signaling via NFκB, yet, the inference of the TNFα pathway activity across major cell types revealed an increased activity in CD14$^+$ monocytes relative to the other cell types (Supplementary Figs. 9b and 8c). In line with the flow cytometry analysis, we detected a strong expansion of myeloid cells, in particular monocytes, with increasing inflammatory stage and a decline of T and B cells in INF$_{high}$ EPTB (Fig. 4e and Supplementary Fig. 9d).

Correlation analysis of measured and estimated cell type frequencies of all datasets (scRNA-seq vs flow cytometry; scRNA-seq vs. bulkRNA-seq and flow cytometry vs. bulkRNA-seq) revealed significant correlations across modalities, highlighting the comparability of the different data types (Supplementary Fig. 10a–c).

Monocytes (20,395 in total) were subclustered into eight distinct populations (Fig. 4f and Supplementary Fig. 11a). Frequencies differed largely across EPTB immunotypes. Patrolling and anti-viral non-classical CD16$^+$ and type I and II IFN positive CD14$^+$IFI$^+$ monocytes peaked in the INF$_{int}$ group. All other pro-inflammatory monocyte populations

(classical CD14$^+$) gradually increased with the EPTB inflammatory staging (Fig. 4g and Supplementary Fig. 11b). The strong complement activation signature observed in bulk transcriptomes was induced by an expansion of a particular cell state of intermediate monocytes (CD14$^+$CD16$^+$C1Q$^+$), which strongly enriched for a signature of C1Q$^+$ intermediate monocytes derived from active TB patients (Supplementary Fig. 11c)[35].

Next, we assessed the enrichment of type I and II IFN across monocyte subsets to identify the cellular drivers of IFN signaling characterizing EPTB. Compared to healthy controls, we observed increased type I IFN signaling within all 8 monocyte cell states, peaking in INF$_{int}$ EPTB and lowest enrichment in INF$_{high}$ EPTB (Fig. 4h). In contrast, type II IFN-associated genes gradually increased and were expressed at similar levels in both INF$_{int}$ and INF$_{high}$ groups across all monocyte populations (Supplementary Fig. 11d).

The CD14$^+$IFI$^+$ cell state exhibited the most significant changes in both type I and II IFN pathways and, within INF$_{low}$ and INF$_{int}$ EPTB, exhibited the highest enrichment for the type I IFNs (Fig. 4h and Supplementary Fig. 11d). Surprisingly, in cells derived from INF$_{high}$ EPTB patients, type I IFN signaling dropped below the level of cells derived from INF$_{low}$ EPTB patients, but was still elevated when compared to healthy controls (Fig. 4h). DE analysis within CD14$^+$IFI$^+$ monocytes exhibited a significant number of DEGs peaking in INF$_{int}$ EPTB (Fig. 4i). In addition to type I IFN signaling in INF$_{int}$ and INF$_{high}$ patients, we found genes associated with IL-1 responses, a surrogate pro-inflammatory pathway, enriched in the CD14$^+$IFI$^+$ population in INF$_{high}$ EPTB (Fig. 4j, Supplementary Fig. 11e and Supplementary Data 3). Interestingly, in INF$_{int}$ EPTB, the immune response started to switch from type I IFN to IL-1 and inflammasome signaling (IL-1 production; *NLRP3* and *CASP1*) (Fig. 4j, k and Supplementary Fig. 11f–h), which changed in INF$_{high}$ EPTB with enrichment primarily for IL-1 response genes (*CCL3* and *CCR1*) (Fig. 4k and Supplementary Fig. 11g). Module score analysis confirmed the modular immune responses of type I IFN and IL-1 signaling within the CD14$^+$IFI$^+$ population depending on EPTB immunotype (Supplementary Fig. 11i–k). Together, by focussing on the specific population of CD14$^+$IFI$^+$ monocytes, we were able to reveal that this cell state undergoes a strong and highly dynamic phenotypic shift involving at least two major innate immune signaling pathways.

## EPTB induces a cytotoxic T and NK cell phenotype switch in conjunction with the appearance of a unique activated CD4 memory T cell state

To dissect their immune response at single-cell resolution, we subclustered 29,662 T and NK cells, resulting in the annotation of 13 distinct cell populations (Fig. 5a, Supplementary Fig. 12a and Supplementary Data 2). Across EPTB immunotypes, the CD8$^+$ T cell compartment depicted increased frequencies of *GZMH*-positive effector memory cells (*GZMH*$^+$ CD8$^+$ T$_m$) while naïve cell frequencies (CD8$^+$ naïve) were reduced. In addition, an activated T cell subset (aCD4T$_m$) that was absent in controls emerged within the CD4$^+$ memory T cell compartment in EPTB patients, particularly in the INF$_{high}$ immunotype. In line with the flow cytometry analysis, NK cell

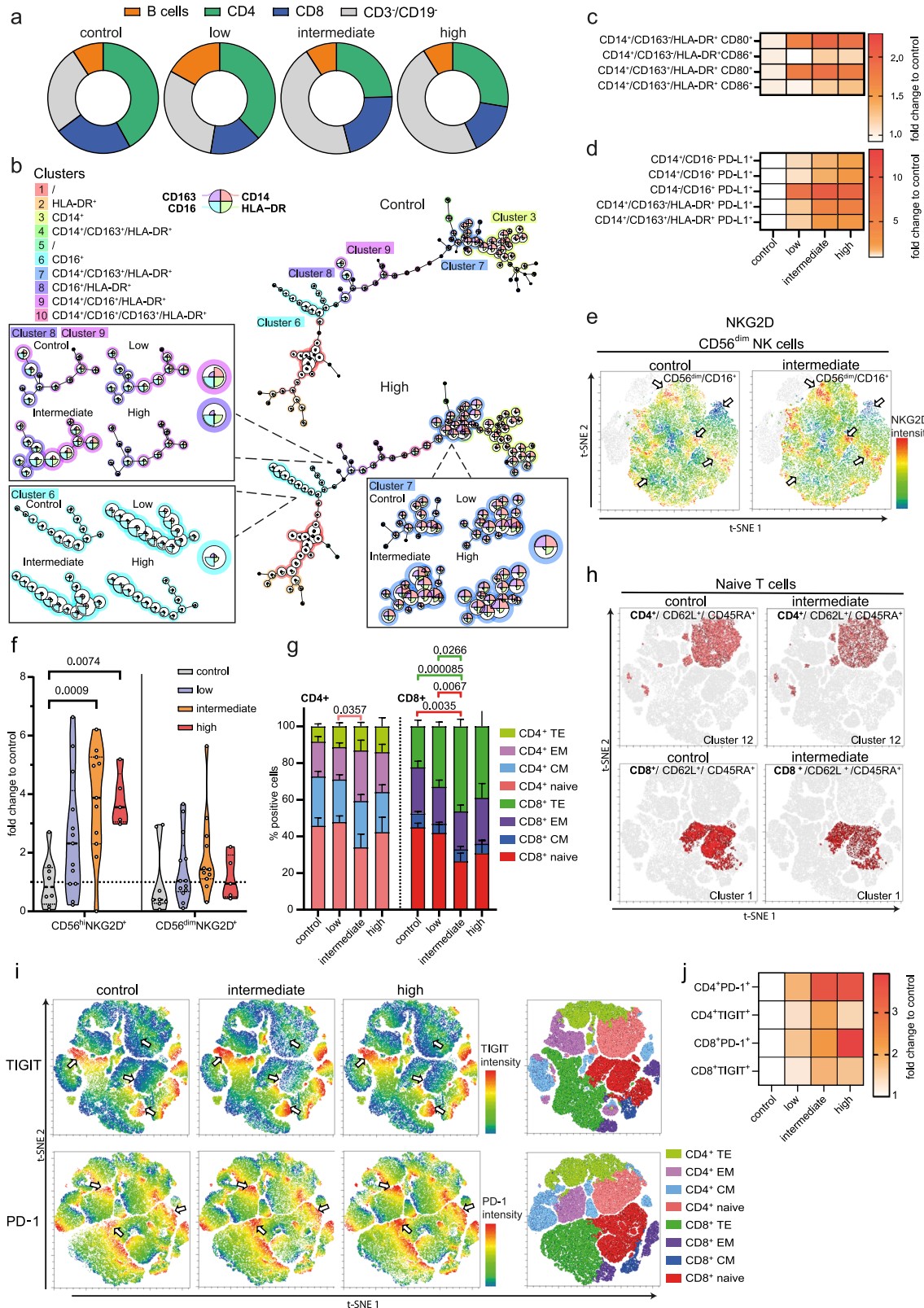

frequencies were most prominently elevated in the INF$_{int}$ immunotype (Fig. 5b and Supplementary Fig. 12b).

Next, we assessed the gene expression landscape of T and NK cell subsets to unveil transcriptional alterations in EPTB with a focus on characterizing the compartments with expanded cell states. The DE analysis revealed an increasing number of uniquely upregulated DEGs with increasing inflammatory stage in both the CD4$^+$ and CD8$^+$ memory

T cell compartments. Notably, CD56$^{dim}$ NK cells exhibited a peak in DEG counts at the INF$_{int}$ stage, mirroring their frequencies (Fig. 5c and Supplementary Fig. 12c). Enrichment analyses of the upregulated DEGs in EPTB highlighted an increasing number of genes associated with the adaptive immune response, T cell activation, cytotoxicity, as well as IFN response pathways towards the INF$_{high}$ immunotype in memory T cells (Fig. 5d, Extended Fig. 12d, e). These trends were confirmed by

**Fig. 3 | Multi-dimensional flow cytometry analysis characterizes EPTB immunotypes. a** Changes in PBMC-subtype ratios in EPTB patients compared to healthy controls. **b** FlowSOM minimum spanning trees displaying changes of myeloid subtypes in healthy controls and $INF_{high}$ EPTB patients. Clustering of $CD3^-/CD19^-/CD56^-/CD123^-$ cells based on CD14, CD16, CD163, and HLA-DR expression. Clusters of interest are magnified and displayed for healthy controls and all groups of EPTB patients. Pie charts show the representative marker expression within clusters. **c** Heatmap of CD80/86 expression in $CD14^+/CD163^+/HLA-DR^+$ and $CD14^+/CD163^-/HLA-DR$ cells. Fold expression to control depicted. **d** Heatmap of PD-L1 expression in myeloid subtypes. Fold expression to control depicted. **e** t-SNE of $CD56^+$ cells displaying relative NKG2D expression in $CD56^{dim}$ NK cells of healthy controls and $INF_{int}$ EPTB patients. t-SNE calculated based on NKP44, NKG2D, CD94, CD16 and CD56 expression. Arrows point towards representative areas with visible changes. **f** Violin plot of $NKG2D^+$ cells in $CD56^{hi}$ and $CD56^{dim}$ NK cells. Fold changes to the mean control value are plotted. Statistics are measured with a two-way-ANOVA with Turkey's post-test. **g** Stacked barchart of the T- cell subtype distribution in T cells of healthy controls and EPTB patients. Statistics are indicated with the respective colors and measured with a two-way-ANOVA with Turkey's post-test. Error bars show SEM. Control $n = 8$ donors, low $n = 13$ donors, intermediate $n = 11$ donors, high $n = 5$ donors (**h**) t-SNE plots highlighting the proportion of naive T cells in healthy controls and $INF_{int}$ EPTB patients (Cluster 1 and 12). T-SNE and FlowSOM are based on CD4, CD8, CD62L, CD45RA, PD-1 and TIGIT expression. **i** T-SNE displaying relative TIGIT and PD-1 expression in T cells of healthy controls, $INF_{int}$ and $INF_{high}$ EPTB patients. PD-1 t-SNE (bottom) calculated based on CD4, CD8, CD62L, CD45RA and PD-1 expression. Arrows point towards representative areas with visible changes. The merged $PD-1/TIGIT^+$ and $PD-1/TIGIT^-$ FlowSOM clusters for the respective T cell subtypes are projected onto the t-SNE in different colors. **j** Heatmap (fold change to control) of PD-1 and TIGIT expression in $CD4^+$ and $CD8^+$ T cells. Source data are provided as a Source Data file for the respective figures.

signature enrichment analyses of the respective pathways' gene sets (Fig. 5e). In $CD56^{dim}$ NK cells the highest count of DEGs related to the aforementioned pathways was detected in the $INF_{int}$ immunotype, which was consistent with the hCoCena analysis. Signature enrichment levels were comparable between the $INF_{int}$ and $INF_{high}$ immunotype (Fig. 5d, e and Supplementary Fig. 12d). In addition to the upregulated gene sets, we also identified a conserved pattern of downregulation across EPTB immunotypes relative to controls. These downregulated genes were associated with various cellular processes and responses (Fig. 5c and Supplementary Fig. 12c, f).

Deconvolution on the cell state level revealed $GZMH^+$ $CD8^+$ $T_m$ and $aCD4T_m$ as major contributors to the observed cellular phenotype of T cells in EPTB, defined by cytotoxicity, cell activation and IFN response (Fig. 5f). The $aCD4T_m$ state was additionally characterized by an enrichment of genes related to TNFα-pathway activity in line with the previous cytokine measurements in the high inflammation stage (Fig. 5g).

To understand whether the rare $aCD4T_m$ cell state, which was almost exclusively found in $INF_{high}$ EPTB patients, is indeed EPTB-specific, we assessed its presence within three independent cohorts of COVID-19, Influenza, and lung cancer patients (Supplementary Fig. 13a). Cell type label transfer was used for the annotation of 179,910 T and NK cells across cohorts (Fig. 5h) and revealed that $aCD4T_m$ cells were mostly absent across the other infectious disease conditions (Supplementary Fig. 13b). In line, the top marker genes of $aCD4T_m$, such as *AQP3*, which is associated with T cell migration, as well as *DPP4* and *ANXA5*, as markers of T cell activation and co-stimulation, showed significantly higher expression in EPTB compared to the other disease groups (Fig. 5i). In contrast to $aCD4T_m$, most of the top marker genes found in the EPTB-associated $GZMH^+$ $CD8^+$ $T_m$ state were expressed at similar levels across the other disease cohorts indicating a common role for this cell state across diseases (Supplementary Fig. 13c). Together, our data deciphered a general switch in T and NK cell states towards stronger cell activation, IFN signaling and cytotoxicity in EPTB as well as the appearance of a unique activated $CD4^+$ T cell state.

## Differential gene expression in $CD14^+$ monocytes contributes to a whole blood-based signature for improved TB diagnostics

Several whole blood gene expression signatures have been proposed as PTB biomarkers[7–14]. Despite the development of a prototype commercial test, consensus on optimal diagnostic gene sets remains lacking[27].

To extend recent developments in PTB to EPTB, we aimed at identifying a diagnostic signature with high accuracy in the detection of EPTB patients across a large spectrum of clinical presentations (Fig. 6a). First, we derived a 15 gene EPTB signature (EPTB-core) defined by the constitutive differential regulation of these genes in comparison to healthy controls independent of the EPTB immunotype (intersection in Fig. 6a and Supplementary Table 2), achieving an AUC value of 0.9662 when delineating healthy individuals from EPTB patients (Fig. 6b). Further, the EPTB-core outperformed other random signatures of the same size following a gene set variation analysis between healthy individuals and EPTB patients (Supplementary Fig. 14a). Significant differences of the EPTB-core gene signature were observed independent of the immunotypes we detected (Fig. 6c). While four of the EPTB-core genes are part of published diagnostic signatures[7–14,16], all other genes are unique to the EPTB-core (Supplementary Fig. 15). Using our scRNA-seq data we next aimed to identify the cell state primarily contributing to the EPTB-core, revealing that most genes were highly expressed in $CD14^+$ monocytes (Supplementary Fig. 14b) and in particular within the $CD14^+IFI^+$ monocyte cell state (Fig. 6d).

Blood-based tests to stage disease severities have become an important research focus in many disease areas, such as cancer or auto-immune diseases. In TB, this approach may allow for personalized HDTs taking different immunotypes into account. In order to efficiently discriminate the three EPTB groups, we developed group-specific gene signatures for $INF_{int}$ and $INF_{high}$ EPTB using a GSVA-based gene prioritization approach of DEGs identified between the EPTB group of interest (see Methods 'GSVA-based gene prioritization approach', Supplementary Fig. 14c). Interestingly, this approach revealed that the addition of 3 DEGs for $INF_{int}$ EPTB and 6 DEGs for $INF_{high}$ EPTB lead to significant discrimination of EPTB groups (Fig. 6e and Supplementary Table 2), while outperforming random signatures of the same size (Supplementary Fig. 14d, e).

Next, we confirmed that the 15-gene EPTB-core effectively discriminates EPTB from a large series of other diseases, namely sarcoidosis, COVID-19, bacterial infections, sepsis, septic shock, influenza, pneumonia or lung cancer by using published transcriptome datasets[36–39] (Fig. 6f and Supplementary Fig. 14f). Following the integration of the datasets, PCA revealed distinct differences in transcriptional profiles when compared to other diseases (Fig. 6g). The subsequent GSVA enrichment of the EPTB-core across the different diseases highlighted the specificity of the signature for EPTB (Fig. 6h). To compare the performance of the EPTB-core signature to previously published PTB signatures, we combined common classification performance evaluation metrics, such as AUROC, precision, recall, and specificity into a comprehensive performance score. Even though some of the other signatures performed better in certain metrics, the EPTB-core offered the best trade-off across all metrics reaching the highest overall performance score (3.55) in comparison to nine recently published signatures (Fig. 6i). Finally, we asked if the EPTB-core was capable of distinguishing all forms of TB (EPTB and PTB) from other diseases using the same evaluation. Once again, the highest performance score (3.458 overall score) was achieved by our EPTB-

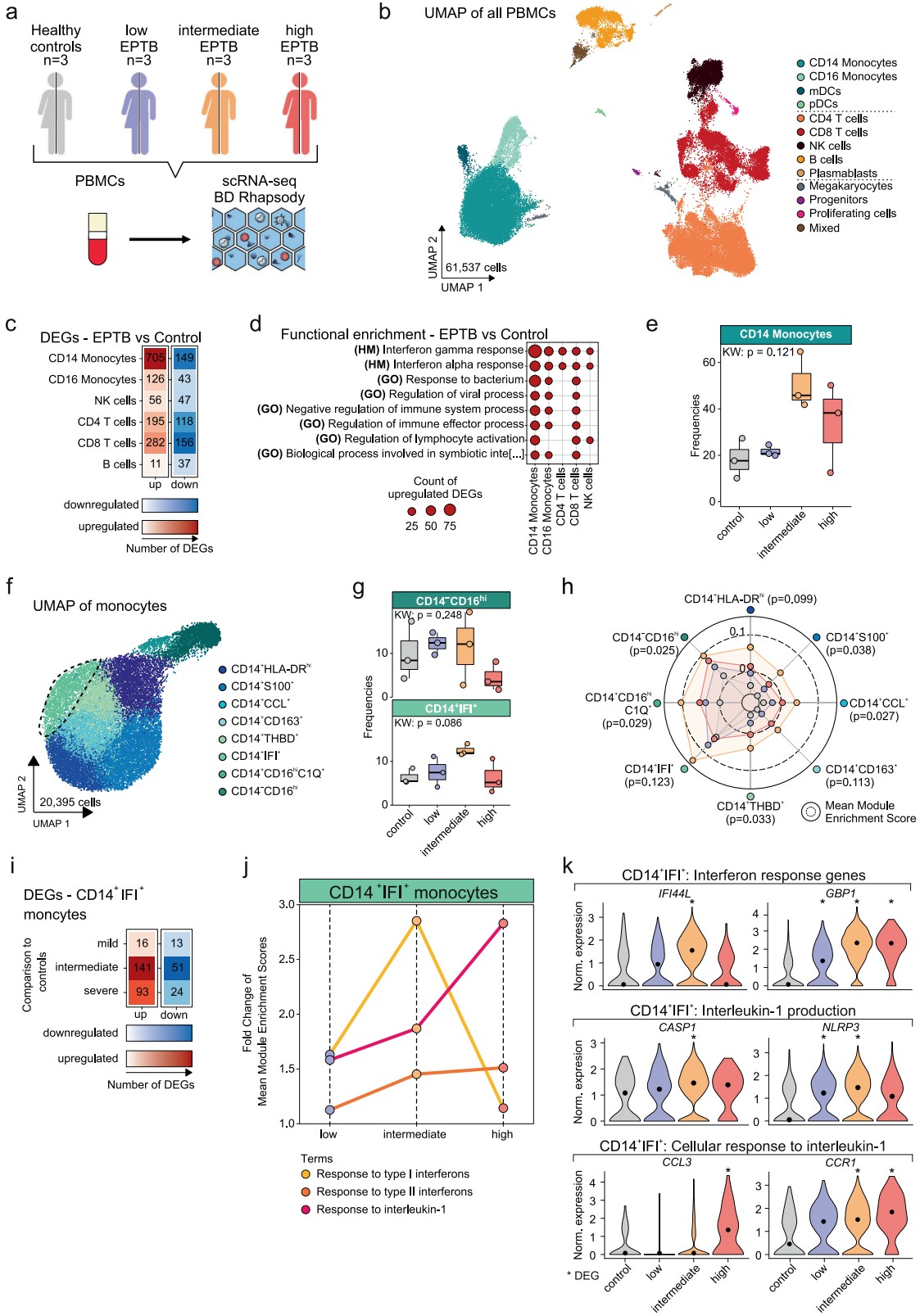

core signature, highlighting its potential applicability not only for the efficient detection of EPTB but also for identifying TB regardless of the infected organ (Supplementary Fig. 14g).

## Discussion

In this study, we performed deep immune profiling of a clinically well-characterized cohort of EPTB patients with highly heterogeneous disease. We hypothesized that a data-driven, omics-based staging of EPTB could provide valuable insights into the pathogenicity of *Mtb*.

Interestingly, our global whole-blood based transcriptomic analysis confirmed this hypothesis and revealed three clearly distinct groups (immunotypes) of EPTB disease stages, each characterized by unique immunological features (Fig. 7). The levels of routine diagnostic inflammatory markers in blood correlated significantly with

**Fig. 4 | Single-cell RNA-sequencing of monocytes. a** Schematic of experimental workflow of scRNA-seq. Created in BioRender. Dahm, K. (2025) https://BioRender.com/8pku9jn. **b** UMAP of the PBMC space ($n = 61,537$) from scRNA-seq, annotated and colored by cell type. **c** Heatmap of the number of DEGs between all EPTB patients and healthy controls per cell type, with more than 2000 cells colored by direction of regulation and number of DEGs. **d** Dot plot of functional enrichment of upregulated DEGs in cell types with more than 2000 cells identified in EPTB patients using the GO biological processes, KEGG, and MSigDB Hallmark gene set databases. Only top terms ordered by Benjamini-Hochberg-adjusted $p$-value significantly enriched in at least three comparisons ($p$-value < 0.05) are displayed. **e** Boxplot of percentage of cells for controls and EPTB immunotypes of CD14 monocytes. Boxplots show the 25%, 50% (median) and 75% percentile, whiskers denote 1.5 times the interquartile range (applies to all following boxplots). Statistics were computed by the Kruskal-Wallis-test. For each group: $n = 3$ donors. **f** UMAP of all monocytes ($n = 20,395$) colored by cell state. Dashed line highlights the CD14$^+$IFI$^+$ population. **g** Boxplot of percentage of cells for controls and EPTB patients of CD14$^+$IFI$^+$ and CD14$^-$CD16$^{hi}$ monocytes. Statistics were computed by the Kruskal-Wallis-test. For each group: $n = 3$ donors. **h** Radar plot of the median module enrichment scores (ES) of the GO term 'response to type I IFN' per monocyte state. Dots represent the median ES per immunotype. Statistics were computed with a Kruskal-Wallis-test. **i** Heatmap of the number of DEGs between EPTB immunotypes and healthy controls within the CD14$^+$IFI$^+$ monocytes, colored by direction of regulation and number of DEGs. **j** Line plot of median module ES fold changes of the GO terms 'response to type I IFN', 'response to type II IFN', 'cellular response to interleukin-1' and 'response to interleukin-1' between EPTB immunotypes and healthy controls in CD14$^+$IFI$^+$ monocytes. Dots represent the mean enrichment of donors. **k** Violin Plot of IFN response genes, IL1 production genes and IL1 response genes in CD14$^+$IFI$^+$ monocytes. Asterisks indicate DEGs between the EPTB immunotypes and healthy controls. Source data are provided as a Source Data file for the respective figures.

these immunotypes, validating our approach. Furthermore, imaging-based quantification of affected body sites mirrored the INF$_{low}$ and INF$_{high}$ immunotypes, with predominantly cervical lymph node EPTB found in INF$_{low}$ and disseminated EPTB in INF$_{high}$ patients. As expected, the INF$_{int}$ patient group showed some overlap across both ends of the clinical spectrum, with a predominant involvement of multiple lymph nodes.

Despite microbiologically confirmed active TB in INF$_{low}$ EPTB, few genes were differentially regulated in whole blood. Bulk and scRNA-seq analyses showed immunotypes with upregulated type I and II IFN signaling and slight expansion of monocytes. In INF$_{int}$ EPTB, NK and T cell-associated genes were strongly upregulated, correlating with increased NK/T cell activation and a shift towards cytotoxicity and pronounced IFN response signaling. Finally, in INF$_{high}$ EPTB, the immunotype changed to massive inflammation, a neutrophil-driven signature, upregulated inflammasome/PRR/DAMP-associated genes and hyperactivated T cells with terminal differentiation. With an excess of cytokines and chemokines in plasma, almost resembling a cytokine storm, INF$_{high}$ EPTB seemed to be characterized by a largely dysfunctional immune response.

Of note, extensive neutrophil-associated signaling was completely missing in the blood of both INF$_{low}$ and INF$_{int}$ patient groups, indicating that in EPTB, only a relatively small fraction of patients is affected by harmful neutrophil related pathogenicity described in most studies focussing on PTB[15,40].

IFN-related immune signatures are a hallmark of PTB, and the extend of pulmonary infiltrates correlates with the abundance of IFN-inducible transcriptional signatures in whole blood[18]. Applying single-cell transcriptomics across distinct groups of EPTB patients, we were able to reveal a more dynamic and fine-tuned alteration of IFN signaling in specific monocyte subsets. CD14$^+$IFI$^+$ monocytes, which predominantly express type I IFN-associated genes in the INF$_{low}$ and INF$_{int}$ groups, appear to be on a pathway toward a more pro-inflammatory state characterized by enhanced expression of inflammasome-related genes and heightened IL-1 responsiveness in INF$_{high}$ EPTB. Thus, our human single cell-derived data provide a strong hint towards a connection between type I IFN and IL-1 signaling as previously observed in *Mtb*-infected mice on a functional level[17,19].

Using scRNA-seq we were also able to identify an EPTB-driven CD4$^+$ T cell subset (aCD4T$_m$). Notably, aCD4T$_m$ cells displayed a distinct immunological phenotype and the corresponding gene expression profile was not found in controls and other infectious diseases. Further studies are needed to decipher the exact immunological function of aCD4T$_m$ in EPTB and whether aCD4T$_m$ are EPTB or even TB-specific. Overall, the T and NK cell phenotypes we describe significantly extend the existing quantitative studies conducted on whole blood samples from TB patients[23,41] and available mouse data[18].

Another study assessing whole blood transcriptional profiles in EPTB patients found correlations between innate immune-associated signatures and disease symptom status but not with the localization or extent of affected body sites[16]. However, this study generated gene expression signatures from microarray data, which are inherently biased compared to RNA-seq. Therefore, methodological differences may explain the varying findings. Notably, both single-cell-based methods (FACS and scRNA-seq) confirmed a strong correlation between immunotypes, blood derived inflammatory markers and the location of affected body sites in our patients.

A study by Wang et al. examining PBMCs of patients with active PTB of varying severity at single-cell resolution, also reported an increase in inflammatory monocytes, whereas lymphocytes decreased in patients with severe manifestations of the disease, which was characterized by the presence of highly cytotoxic T and NK cells[42]. In contrast to our study, the monocyte C1 signature was present in CD16$^+$ cells, and there was a lack of IFN signatures in CD14$^+$ monocyte clusters. These findings point towards both shared and distinct regulatory mechanisms between PTB and EPTB, warranting further investigation.

Our data may also impact future diagnostics of TB, which remains challenging and imprecise. A recently published prospective study exploring the value of a highly reduced whole blood gene expression signature quantifying three genes with the commercially available GeneXpert platform, revealed a relatively low sensitivity of 56% in PTB and even lower values in EPTB[27]. Low sensitivity was also observed in a recent case-control study in which PTB patients were compared to patients suffering from other respiratory diseases[43]. By integrating a broad spectrum of disease stages, we were able to identify a set of 15 DEGs (EPTB-core) which discriminated both EPTB and PTB patients from healthy controls and a series of other diseases better than recently published signatures, including commercially used ones. Importantly, EPTB-core was able to distinguish EPTB from sarcoidosis, a granulomatous non-infectious disease which shows considerable transcriptional overlap with TB and especially EPTB[16]. The diagnostic performance of this signature can now be assessed in larger TB cohorts and more comprehensive analyses using training, test and validation data-sets. In a second step, the signature can be evaluated in prospective clinical trials aiming at improved TB diagnostics. Interestingly, we were able to link whole blood-based data to single-cell transcriptomics of PBMCs, which revealed that CD14$^+$ monocytes, and in particular CD14$^+$IFI$^+$ monocytes, represent the primary drivers of the DEGs found in EPTB-core. This finding may allow for cell enrichment procedures focussing on specific monocyte subsets, followed by selective transcriptome analyses which may further increase the sensitivity and specificity of diagnostic signatures.

The extensive immunological profiling we performed offers profound insights into the diversity of the immune response to *Mtb*,

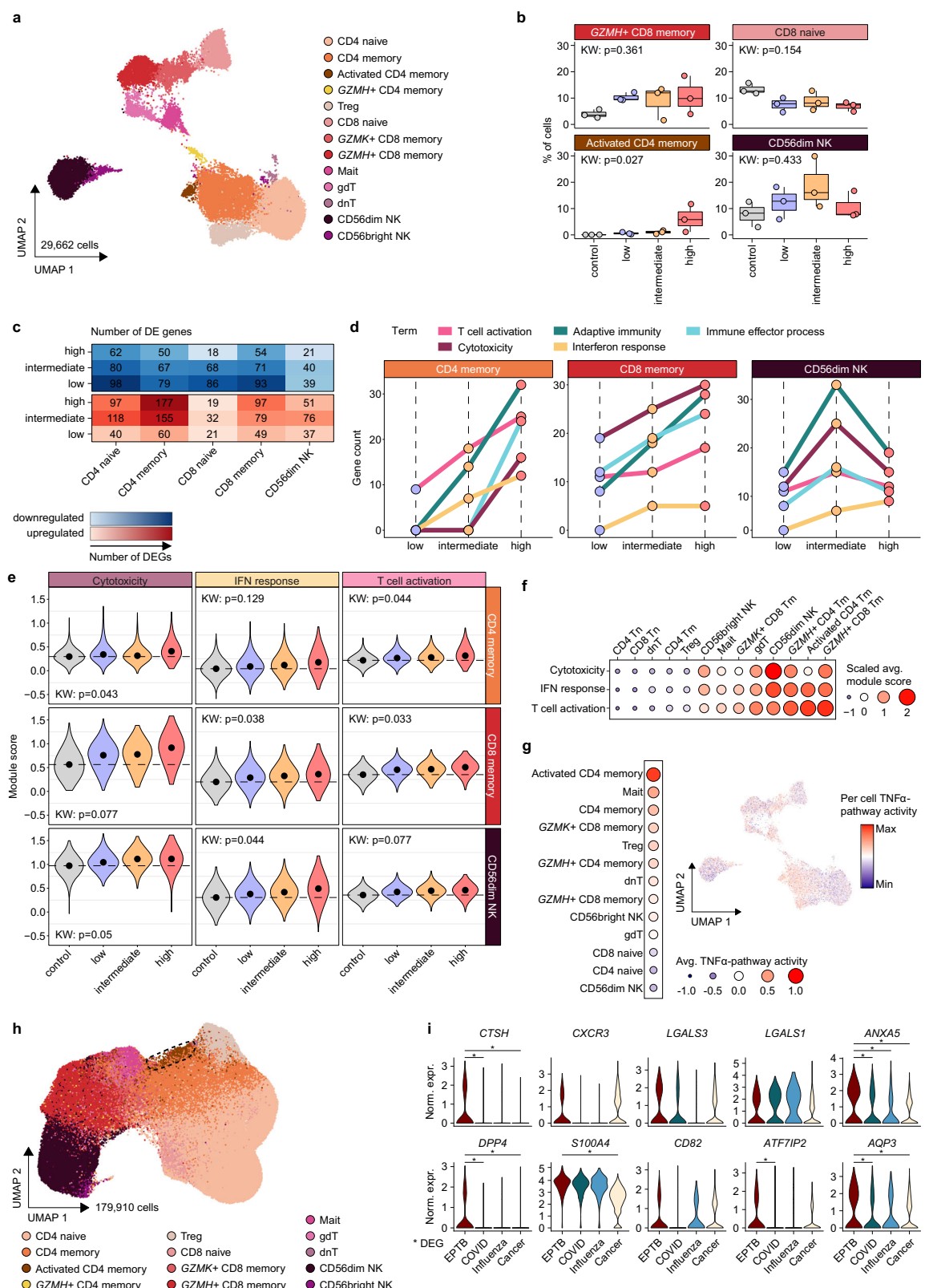

potentially aiding in the identification of patients suitable for more tailored and personalized HDT strategies. According to our data and the complex immunotypes identified on both bulk and single cell level, we assume that a generalized HDT approach may fail in future clinical studies. Staging based on gene expression signatures delineating immunotypes, as performed in our study, may overcome these limitations in clinical trials. Patients presenting with a hyperinflammatory phenotype that seems to be associated with inflammasome activation and IL-1 signaling may benefit from immune-modulatory interventions dampening this response. However, this may not be the case for patients with milder forms of the disease. It is important to mention that our findings also need to be aligned with the local immune response in infected tissue, which represents an important task for the future.

**Fig. 5 | scRNA-seq reveals a T and NK cell phenotypic switch in EPTB. a** UMAP of the T and NK cell space (n = 29,662 cells) from scRNA-seq data colored by indicated cell states (Mait = Mucosal-associated invariant T cells, gdT = gamma-delta T cells, dnT = CD4⁻ CD8⁻ double-negative T cells). **b** Boxplot of percentage of cells for controls and EPTB patients per selected cell state. Boxplots show the 25%, 50% (median) and 75% percentiles, whiskers denote 1.5 times the interquartile range. Statistics were computed by the Kruskal-Wallis test. For each group: n = 3 donors. **c** Heatmap of the number of DEG between EPTB patients and controls per cell subtype with more than 2000 cells colored by direction of regulation and number of DEGs. Cell states were summarized into subtypes of ≥100 cells per condition and cell subtype for differential expression analysis. **d** Summarized functional categories in memory T cells and CD56$^{dim}$ NK cells from enrichment of upregulated genes identified in EPTB patients using the GO biological processes database. Terms were significantly enriched in at least three comparisons (Benjamini-Hochberg-adjusted p-value < 0.05). For each group: n = 3 donors. **e** Violin plot of the ES of selected and summarized GO and KEGG database terms using

upregulated DEGs in EPTB patients. Median ES are indicated as black dots. Statistics were computed with a Kruskal-Wallis-test on the donor level. **f** Median ES per cell state of the signatures from (**e**). Dots are scaled and colored by score. **g** UMAP visualization of the T and NK cell space colored by PROGENy-inferred TNF-α pathway activity score. Median activity scores per cell state are indicated as dots scaled and colored by score. **h** UMAP of the integrated T and NK cell space (n = 179,910 cells) from scRNA-seq data of the EPTB cohort and different disease cohorts from Schulte-Schrepping et al.[32], Zhang et al.[78], and Yanagihara et al.[79] colored by indicated cell states. Cell states are defined by Seurat reference mapping label transfer from the EPTB dataset per cohort. Dashed line highlights the activated CD4⁺ memory cell (aCD4T$_m$) population. **i** Violin plots of the top marker genes of aCD4T$_m$ in EPTB and other disease cohorts. aCD4T$_m$ are defined by Seurat reference mapping label transfer from the EPTB dataset. Significantly higher expression in EPTB is indicated with asterisks (parameters: log2FC = 0.25, min.pct = 0.1, unpaired two-sided Wilcoxon test, Bonferroni-adjusted p-value < 0.05). Source data are provided as a Source Data file for the respective figures.

To conclude, our extensive whole blood- and single cell-based analysis precisely identified highly heterogeneous groups of EPTB patients, each characterized by distinct clinical and immunological features. Together, the data reveal dynamic trajectories of innate and adaptive immune responses in different stages of EPTB, which exacerbate towards a hyperactivated T cell response with monocyte/neutrophil-driven hyperinflammation linked to IFN and IL-1 signaling in severe forms of the disease. These findings can now be further exploited for improved diagnostics and therapeutics in TB. This study provides a detailed immunological characterization of a unique cohort of EPTB patients, leading to the development of a highly accurate diagnostic gene signature. Some limitations apply. We are currently lacking a comprehensive understanding of the interplay between innate and adaptive immune components in TB. Our bulk and single-cell data suggest complex interactions between these immune cell populations, warranting further investigation. In addition, it is important to note that our cohort does not include patients with *Mtb* infection of the central nervous system (CNS). This unique TB manifestation, in which mycobacteria interact closely with CNS-derived immune cells, requires dedicated studies in regions with higher prevalence. Nevertheless, the value of our blood-based signatures can be readily assessed in clinical cohorts focusing on CNS-TB. Further, our EPTB-core gene expression signatures need to be comprehensively evaluated in larger TB cohorts, preferably multicentric cohorts, and with the usage of training, test and validation datasets.

## Methods
### Study design and patient cohort
Patients were enrolled between August 2018 and September 2021 at the University Hospital Cologne, division of infectious diseases. EPTB diagnosis was made in patients with the detection of *M. tuberculosis* complex (culture positivity and/or PCR positivity) from bodily secretions or tissue. In three culture-negative patients, a clinical diagnosis was made based on epidemiological exposure combined with physical findings, radiographic findings, and appropriate histopathological findings (e.g., necrotizing granuloma, acid-fast staining positivity). Except for one patient with missing data, all patients showed positive results in TB-specific IFN gamma release assays (IGRA), confirming previous infection with *M. tuberculosis* complex. All patients underwent systematic baseline imaging, including chest X-ray, abdominal ultrasound, and lymph node ultrasound. For specific clinical findings, suspected disseminated disease, or inconclusive chest X-ray results, additional computed tomography (CT) or magnetic resonance imaging (MRI) was performed to assess the extent of the disease within a staging protocol. Radiological data were centrally reviewed to document organ involvement, lesion characteristics, and evidence of dissemination, defined as involvement of two or more non-contiguous

sites. Baseline assessments included symptom history, physical examination, and measurement of inflammatory markers such as CRP, leukocyte count, and ESR. Patients suffering exclusively from central nervous system (CNS) TB were excluded from the study, as CNS TB primarily affects the neuroimmune system.

Detailed information on all patients included in the study are provided in Supplementary Table 1. Sample size and epidemiological imbalances of EPTB did not allow for matching according to the sex of patients. IGRA negative healthy volunteers were used as controls.

### Peripheral blood mononuclear cell (PBMC), plasma and serum isolation of whole blood
PBMCs were purified using density gradient centrifugation using Cytiva Ficoll-Paque® (Fresenius, Bad Homburg, Germany) and Leucosep™ tubes (Greiner, Kremsmünster, Austria). At this step plasma was taken off and stored at −80 °C. Afterwards, PBMCs were washed several times with PBS and stored in freezing media (FBS + 10% DMSO) at −150 °C.

### Whole blood RNA isolation
PAXgene Blood RNA Kit (Preanalytix, Hombrechtikon, Germany) was used according to the manual provided by the vendor. Briefly, PAX tubes were thawed at room temperature, centrifuged and washed with RNase-Free Water. Next, pellet is resuspended in resuspension buffer and binding buffer and proteinkinase K. Incubation times were performed as recommended and the spin column procedure was performed as suggested by the manual. Upon elution, RNA was incubated for 5 min at 65 °C and RNA was stored at −80 °C.

### Bulk RNA sequencing
Sequencing library prep was performed with in total input of 100 ng RNA according to the NEBNext Ultra RNA library prep protocol (New England Biolabs, Ipswich, MA, USA). Next, the RNA libraries were validated with Tape Station 4200 (Agilent Technologies, Santa Clara, CA, USA). Library quantification was performed using the KAPA Library Quantification Kit (VWR International, Radnor, PA, USA) and the 7900HT Sequence Detection System (Applied Biosystems, Foster City, CA, USA). Sequencing was done with NovaSeq6000 sequencers (Illumina, San Diego, CA, USA) with a PE100bp read length, aiming at 50 M reads/sample. Quality control of samples was performed using fastQC (v0.11.9) (https://www.bioinformatics.babraham.ac.uk/projects/fastqc/) and multiQC[44] (v1.14) (https://multiqc.info/). Sequenced reads were aligned against the Gencode human reference genome v33 using STAR[45] (v2.7.10b) (https://github.com/alexdobin/STAR, https://doi.org/10.1093/bioinformatics/bts635) and reads were summarized using no strandness information. The alignment pipeline was executed using SnakeMake (v7.20.0) (https://snakemake.github.io/, https://doi.org/10.12688/f1000research.29032.2).

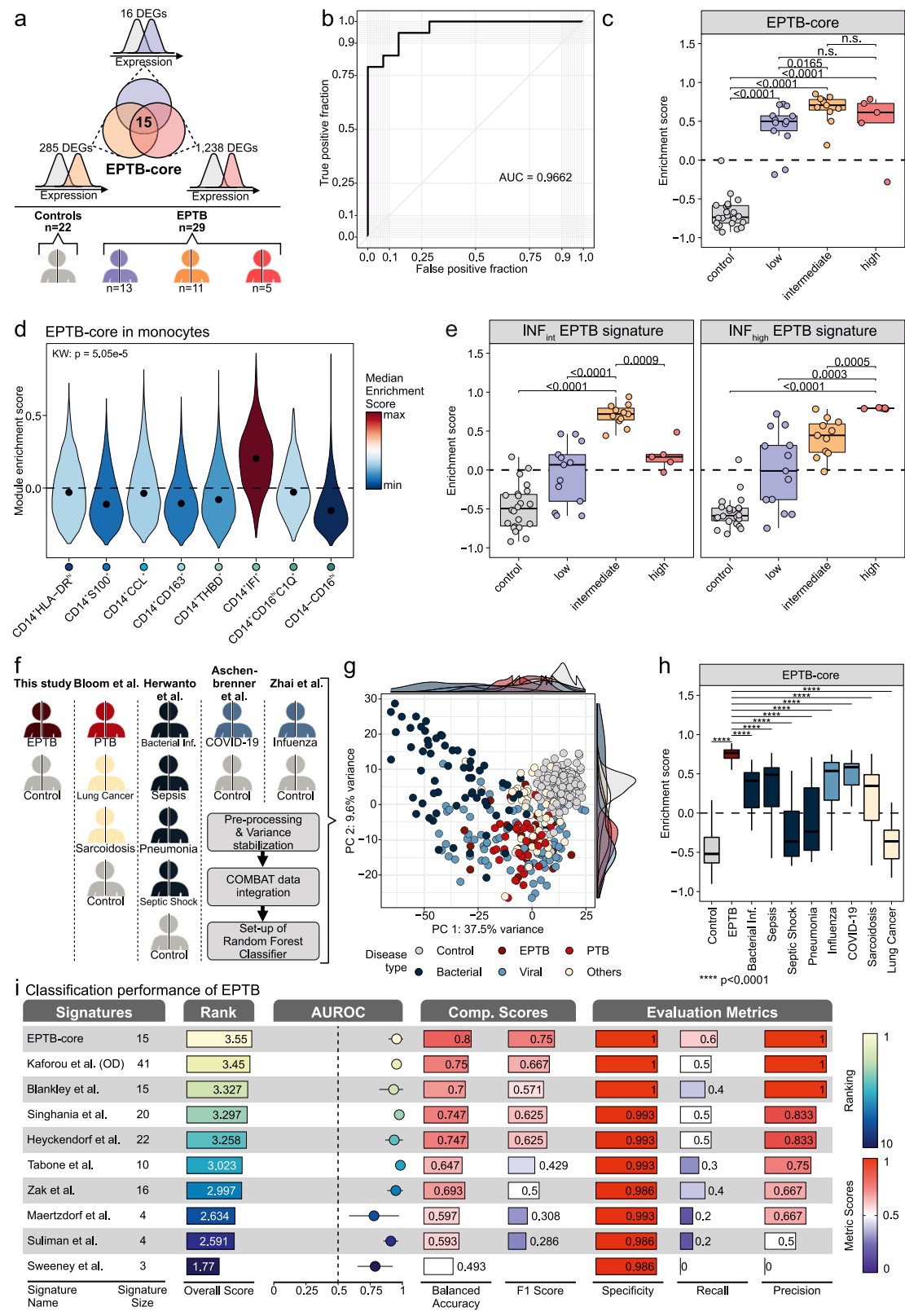

## Bulk RNA-seq data preprocessing

The following analysis steps were performed in R[46] (v4.1.0) and R Studio[47] (v1.4.1717). After importing the count matrix, gene with less than 10 counts in less than 4 samples were excluded, resulting in 20,206 genes kept in the analysis. Normalization of the count matrix was computed with DESeq2[48] (v1.32.0), a variance stabilizing transformation applied using the DESeq2 *vst* function at default settings and

batch-corrected for different sequencing time points with limma[49] (v3.48.3).

## Differential expression analysis in bulkRNA-seq data

Differential expression analysis based on DESeq2 was performed by adjusting *p*-values according to independent hypothesis weighting from IHW[50] (v1.20.0) at default settings and applying empirical Bayes

**Fig. 6 | EPTB-core signature shows high accuracy in the detection of TB.**
**a** Schematic overview of signature detection. Created in BioRender. Dahm, K.
(2025) https://BioRender.com/dis8e5p. **b** Area under the receiver operating characteristic curve (AUROC) of the EPTB-core signature in balanced datasets. **c** Boxplot
of GSVA ES of the EPTB-core split by EPTB immunotype. Boxplots show the 25%,
50% (median) and 75% percentiles, whiskers denote 1.5 times the interquartile range
(applies to all following boxplots). Statistics were computed by an unpaired two-
sided Wilcoxon test followed by Benjamini-Hochberg adjustment. Control $n = 22$
donors, low $n = 13$ donors, intermediate $n = 11$ donors, high $n = 5$ donors. **d** Violin
plot of module ES of the EPTB-core signature in across monocyte states. Statistics
were computed with a Kruskal-Wallis-test on the donor level. **e** Boxplot of GSVA ES
of the $INF_{int}$ and $INF_{high}$ EPTB-specific signatures split by EPTB immunotype. Sta-
tistics were computed by an unpaired two-sided Wilcoxon test follow by Benjamini-
Hochberg adjustment. Control $n = 22$ donors, low $n = 13$ donors, intermediate $n = 11$
donors, high $n = 5$ donors. **f** Schematic overview of the datasets integrated into the

'Other Disease' (OD) datasets. Created in BioRender. Dahm, K. (2025) https://
BioRender.com/ipif5tp. **g** PCA of the 1000 most variable genes of the OD datasets
colored by disease type. **h** Boxplot of GSVA ES of the EPTB-core split by disease and
colored by disease type. Statistics were computed by an unpaired two-sided Wil-
coxon test follow by Benjamini-Hochberg adjustment. Control $n = 229$ donors,
EPTB $n = 29$ donors, Bacterial infection $n = 12$ donors, Sepsis $n = 20$ donors, Septic
Shock $n = 19$ donors, Pneumonia $n = 14$ donors, Influenza $n = 42$ donors, COVID-19
$n = 39$ donors, Sarcoidosis $n = 39$ donors, Lung cancer $n = 16$ donors. **i** Classifier
overview of the EPTB-core and 9 published pulmonary TB (PTB) signatures in
identifying EPTB from other diseases and healthy controls. The first bar plot
represents the overall score of the AUROC, specificity, precision, and recall, colored
by the ranking of the signature. The point range plot depicts the AUROC and the 95
% confidence interval of the respective AUROC. The following bar plots depict the
balanced accuracy, specificity, precision, recall, and F1 score of each signature.
Source data are provided as a Source Data file for the respective figures.

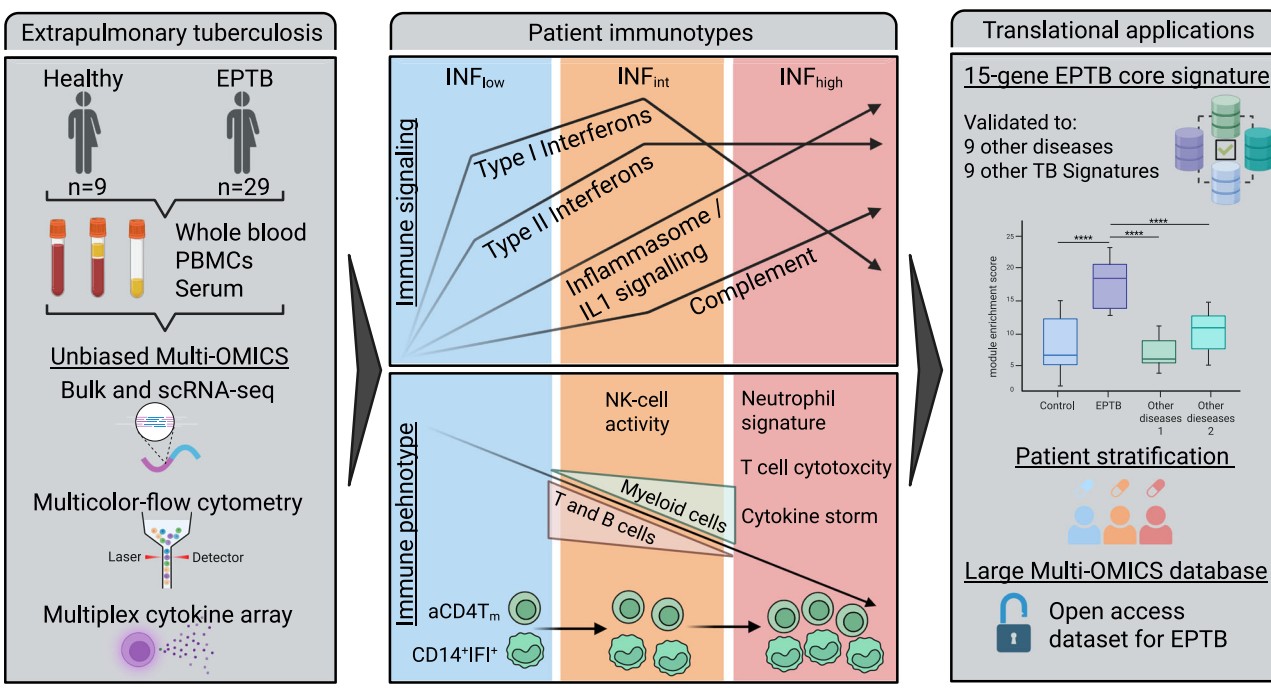

**Fig. 7 | Graphical summary of the key findings made in this study.** Created in BioRender. Lange, D. (2025) https://BioRender.com/rcaya0x.

shrinkage estimators from apeglm[51] (v1.14.0). Genes with an abs(fold
change) > 2 and an adjusted *p*-value < 0.05 were defined as differen-
tially expressed genes.

## Gene set enrichment analysis

Gene Ontology (GO)[52,53], Kyoto Encyclopedia of Genes and Genomes
(KEGG)[54–56] and Molecular Signature Database Hallmark gene set[57,58]
enrichment was performed the clusterProfiler[59] (v4.4.4) *enricher*
function. Gene-Term associations for the respective databases were
downloaded from the Molecular Signature Database[57] (v2023.1).

## Gene co-expression network analysis

Gene co-expression network analysis was performed using hcocena[31,60]
(v1.0.0). The 9010 most variable genes of the normalized, batch-
corrected data were used as input and gene-gene correlations were
computed using Pearson's correlation. Gene pairs with a Pearson's
correlation coefficient lower than 0.751 were excluded from the net-
work, resulting in a network with 5042 genes, 178,105 edges and an $R^2$-
value of 0.8. Leiden clustering with a resolution of 1 identified 10
modules with a minimum size of 50, containing a total of 4965 genes.

## Cell type deconvolution

Cell type abundances of the batch-corrected data were determined by
CIBERSORTx[61] (https://cibersortx.stanford.edu/) with default para-
meters using BD Rhapsody datasets of cohort 2 from
EGAS00001004571[32] filtered for whole-blood samples and healthy
controls as reference. From each cell type, 1000 cells were sampled if
available, and the normalized data was used to generate the single-cell
reference matrix.

## Classifier set up and signature assessment

To create a balanced dataset of control and EPTB patients, 13 addi-
tional control samples were pre-processed. Genes with less than 10
reads in less than 2 samples were excluded, resulting in a count matrix
of 19,979 genes. Further pre-processing steps were performed as
previously described. The combined datasets was filtered for the
intersection of genes included in both datasets, consisting of 18,663
genes from 22 control and 29 EPTB patients and genes were ranked per
patient. To assess the classification performance, a random forest
classifier was constructed with caret[62] (v7.0-1) and randomForest (v4.7-
1.2)[63]. The expression values of the combined datasets were rank

transformed and the data split into a training and test datasets in a 2:1 ratio. RF hyper-parameters were set to default and the 'mtry' parameter was set to $\sqrt[2]{n-1}$ where n is the signature length. The performance was estimated using a leave-one-out cross validation and was evaluated with MLeval (v0.3)[64] and by calculating the area under the receiver operating characteristic curve (AUROC) generated with plotROC[65] (v2.3.1) and pROC (v1.18.8)[66]. Gene set variation analysis was performed with GSVA[67] (v1.40.1) without kernel estimation of the cumulative density function on the ranked combined datasets to assess the classification performance for each inflammatory stage. Significance of the results was calculated with a Wilcoxon test followed by a Benjamini-Hochberg adjustment.

## GSVA-based gene prioritization approach
Group-specific EPTB signatures were generated by either using the intersection of identified DEGs between the different EPTB inflammatory stages and control samples or between a selected EPTB inflammation immunotype and the other inflammation immunotypes following an adapted optimization strategy published in Knoll et al.[68]. In short, the intersection of DEGs was ordered by the (average) log2 fold change, and ties were solved based on mean expression level. An iterative GSVA was then performed, starting with the gene with the highest (average) log2 fold change, adding the gene with the next highest log2 fold change with each iteration. For each iteration, a Wilcoxon test was performed between the different EPTB inflammatory stages and control samples or between a selected EPTB inflammatory stage and the other inflammatory stages, and the average p-value was calculated. Genes which increased the p-value were excluded from the signature. The final signature consisted of the gene set with the lowest p-value.

## Enrichment of signatures in other diseases
To assess the disease specificity of the core signature, bulk RNA-seq and microarray datasets of other viral, bacterial or pulmonary diseases were analysed. Datasets were selected based upon the following criteria: (i) The data was generated from whole blood, (ii) sample collection protocols were as similar as possible to the described above (i.e., blood collection tubes), and (iii) included healthy controls as a reference. The following datasets were considered: Bloom et al.[39] (GSE42834 [https://www.ncbi.nlm.nih.gov/geo/query/acc.cgi?acc=GSE42834]), Zhai et al.[38] (GSE68310 [https://www.ncbi.nlm.nih.gov/geo/query/acc.cgi?acc=GSE68310]), Aschenbrenner et al.[37] (EGAS00001004503 [https://ega-archive.org/studies/EGAS00001004503]) and Herwanto et al.[36] (GSE154918 [https://www.ncbi.nlm.nih.gov/geo/query/acc.cgi?acc=GSE154918]) together with the EPTB datasets. Each datasets was pre-processed as previously described and transformed into a log2-space while genes not included in all studies were excluded from the datasets. The final datasets consisted of 494 samples and 11,778 genes. All datasets were integrated with the sva[69] (v3.40.0) ComBat function prior to performing a classification benchmark. GSVA between the different disease states and controls was performed without kernel estimation of the cumulative density function. Significance of the results was calculated with a Wilcoxon test followed by a Benjamini-Hochberg adjustment. For the EPTB core signature and for each PTB signature, an RF classifier was constructed based on the rank-transformed integrated datasets using the architecture described above. Classification performance was evaluated based on an overall score reflecting the sum of the AUROC, specificity, precision and recall.

## BD Rhapsody blood single-cell RNA-seq and AbSeq
Whole transcriptome analyses and surface protein expression, using the BD Rhapsody Single-Cell Multiomics System (BD, Biosciences) were performed on PBMCs of 3 selected donors of each stratified EPTB group and 3 healthy controls.

Cells from four samples per condition were labeled with sample tags (BD Human Single-Cell Multiplexing Kit) following the manufacturer's protocol. Briefly, a total number of $1 \times 10^6$ cells were resuspended in 90 µl of Stain Buffer (FBS) (BD PharMingen). The sample tags were added to the respective samples, and the samples were incubated for 20 min at room temperature. After incubation, 200 µl stain buffer was added to each sample and centrifuged for 5 min at $300 \times g$ and 4 °C. After washing, cells were resuspended in 500 µl of cold Sample Buffer (BD, Biosciences) and counted using a Neubauer Hemocytometer. Based on the counting, labeled samples were pooled equally.

After pooling, the cell suspension was stained with the BD AbSeq Immune Discovery Panel (BD, Biosciences, Cat#25970) to assess surface marker expression. Reconstitution and cell staining were performed according to the manufacturer's protocol. After staining, the cells were resuspended in 300 µl of cold Sample Buffer, and a second cell counting step was conducted. The concentration was adjusted to achieve 120,000 cells in 1300 µl of cold Sample Buffer. The pooled samples were then split in two for super-loading two BD Rhapsody cartridges, with approximately 60,000 cells each.

Single cells were isolated using Single-Cell Capture and cDNA Synthesis with the BD Rhapsody Express Single-Cell Analysis System according to the manufacturer's recommendations (BD Biosciences). cDNA libraries were prepared using the mRNA Whole Transcriptome Analysis (WTA), AbSeq and Sample Tag Protocol (BD Biosciences). The final libraries were quantified using a Qubit Fluorometer with the Qubit dsDNA HS Kit (ThermoFisher), and the size-distribution was measured using the Agilent high-sensitivity D5000 assay on a TapeStation 4200 system (Agilent Technologies). Sequencing was performed in paired-end mode (R1 85 bp and R2 215) on a NovaSeq 6000 with NovaSeq 6000 S2 Reagent Kit (300 cycles) chemistry. After demultiplexing of bcl files using bcl2fastq2 (v2.20) from Illumina and quality control, a barcode whitelist provided by BD Biosciences was used to filter the paired-end scRNA-seq reads for valid cell barcodes. Cutadapt[70] (v1.16) package (Martin, 2011) was used to trim adapter sequences and to filter reads for a PHRED quality score of 20 or above. Next, STAR[45] (v2.6.1b) was used for alignment against the Gencode v33 human reference genome and Dropseq-tools (v2.0.0) were used to quantify gene expression and collapse to UMI count data (https://github.com/broadinstitute/Drop-seq/). For SampleTag oligo-based demultiplexing of single-cell transcriptomes and subsequent assignment of cell barcodes to their sample of origin, the respective multiplexing tag sequences and AB-seq sequences were added to the reference genome and quantified, as well[71].

## scRNA-seq data pre-processing
The following analysis steps were performed in R[46] (v4.1.0) and R Studio[47] (v1.4.1717). scRNA-seq analysis was performed using Seurat[72,73] (v4.0.4). Cells of each cartridge were demultiplexed in a two-step process. First, cells that were considered as doublets based on the Seurat HTODemux function at default settings were discarded prior to genotype-based demultiplexing using Vireo[74] (v0.5.6) and cellsnp-lite[75] (v1.2.2). Only cells defined as singlets by the Seurat HTODemux function or cells that could be assigned with a 90 % probability to its respective donor based on the single-nucleotide polymorphism (SNP) information were kept. In addition, cells were excluded if they had 1) more than 35 % mitochondrial genes approximately equal to three mean absolute deviations above the median according to scater[76] (v1.20.1), 2) more than 5 % hemoglobin genes, 3) less than 465 and 338 UMIs, as well as 412 and 328 genes based on the inflection point of the cells ranked by either the number of UMIs or genes of each cartridge, respectively, and 4) a prominent multi-lineage marker gene expression. Genes expressed in less than 5 cells were excluded. After QC, a total of 61,537 PBMCs expressing 28,604 genes were analysed.

Normalization, scaling and dimensionality reduction of the WTA and AbSeq data were computed with Seurat functions. The gene expression values were normalized by multiplying the total UMI counts per cell by 10,000 (TP10K) and applying a natural log transformation by log(TP10K + 1). Subsequently, the normalized data was scaled as well as centered and a regression between each gene and donor was performed to account for the intra-individual heterogeneity of each donor. Following a variance stabilization using the 'vst' method, PCA of the 2000 most variable features was performed to reduce the dimensionality of the data. Protein expression data of each cell was normalized by applying a centered log-ratio transformation prior to scaling and centering. A two-dimensional, multimodal representation of the data was calculated by generating a weighted nearest neighbor (WNN) graph of the first 30 PCs of the gene expression data and the normalized protein expression data with the shared nearest neighbor (SNN) pruning cut-off set to 1/25. Nearest neighbors were then used to calculate a UMAP. Cells were clustered based on the WNN graph using the Louvain algorithm at a resolution of 0.3. Cluster identities were determined using the Seurat FindAllMarkers function using the Wilcoxon rank sum test, defining cluster marker genes as genes expressed in at least 25 % of the cells of either of the two compared populations, with a difference of at least 10 % between the populations, and a log fold change > 0.25. During log fold change calculation, the mean was computed as implemented in Seurat[77] (v5):

$$\mathrm{mean}_x = \log_2\left(\left(\sum_{i=1}^{n}(e^{x_i}-1)+1\right)/n\right)$$

where $x$ is defined as the respective gene and $n$ as the number of cells. Subsequently, cells were annotated in a three-step process by overlapping cluster marker genes and protein marker expression with literature-known cell-type marker genes as CD14 Monocytes (*CD14*, *S100A8*, *S100A9*), CD16 Monocytes (CD16, *CDKN1C*, *FCGR3A*), mDCs (*CD1C*, *FCER1A*), pDCs (*CLEC4C*, *IRF7*, *LILRA4*), CD4 T cells (CD3, CD4, *TCF7*), CD8 T cells (CD3, CD8, *CD8A*), NK cells (CD56, *NKG7*, *GZMB*), B cells (*PAX5*, *IGHD*, *CD22*, CD19), Plasmablasts (*CD27*, *IGHA1*, *IGHG1*, *IGHA2*), Megakaryocytes (*TUBB1*, *ITGA2B*, *PPBP*), Progenitors (*CD34*, *GATA2*), Proliferating cells (*TYMS*), and Mixed (CD19, CD14).

## Selection and annotation of monocytes

Monocytes were selected and annotated in an iterative process. Genes expressed in less than 5 cells were excluded from the analysis. In the first iteration, the data was subset to all cells defined as either CD14 monocytes, CD16 monocytes or mDCs. Then, the subset was normalized, scaled and centered as described above. For dimensionality reduction of the gene expression data, a PCA was performed of the 2000 most variable genes following a variance stabilization using the 'vst' method. To create a two-dimensional, multimodal representation of the data, a WNN graph of the first 15 PCs and the normalized and scaled protein expression data was generated with an SNN pruning cut-off set to 1/25. Nearest neighbors were then used to calculate a UMAP. Cells were clustered based on the WNN graph using the Louvain algorithm at a resolution of 0.94. Clusters with high expression of literature-known marker genes of non-monocytes (such as NK cells, T cells or mDCs) in addition to the typical monocyte markers were excluded from further analysis, resulting in 20,395 total monocytes. After the clean-up, a second iteration of the previous pre-processing steps was performed at final clustering resolution of 0.9 and cluster identities were determined using the Seurat FindAllMarkers function using the Wilcoxon rank sum test defining cluster marker genes as genes expressed in at least 25 % of the cells of either of the two compared populations, with a difference of at least 10 % between the populations, a log fold change > 0.25. During log fold change calculation, the mean was computed as described above. Subsequently, cells

were annotated in a two-step process based on cluster marker genes and protein marker expression as CD14⁺HLA-DR^hi (*HLA-DQA1*, *HLA-DPB1*, *HLA-DQB1*), CD14⁺S100⁺ (*S100A12*), CD14⁺CCL⁺ (*CCL3*, *CCL3L3*, *CXCL8*), CD14⁺CD163⁺ (*CD163*, *F13A1*), CD14⁺THBD⁺ (*THBD*, *THBS1*, *RGCC*), CD14⁺IFI⁺ (*IFI44*, *IFIT3*, *IFIT2*), CD14⁺CD16^hiC1Q⁺ (*C1QA*, *C1QB*, *C1QC*), and CD14⁻CD16^hi (*CDKN1C*, *FCGR3A*).

## Selection and annotation of T and NK cells

T and NK cells were selected and annotated in an iterative process. Genes expressed in less than at least 5 cells were excluded. In the first iteration, the data was subset to all cells defined as either CD4 T cells, CD8 T cells or NK cells. Then, the subset was normalized, scaled and centered as described above. For dimensionality reduction of the gene expression data, a PCA was performed of the 2000 most variable genes following a variance stabilization using the 'vst' method. To create a two-dimensional, multimodal representation of the data, a WNN graph of the first 15 PCs and the normalized and scaled protein expression data was generated with an SNN pruning cut-off set to 1/25. Nearest neighbors were then used to calculate a UMAP. Cells were clustered based on the WNN graph using the Louvain algorithm at a resolution of 1.1. Clusters with high expression of literature-known marker genes of non-T/ NK cells or multiple cell types were excluded from further analysis, resulting in 29,662 total T and NK cells. After the clean-up, a second iteration of the previous pre-processing steps was performed with a final clustering resolution of 0.7 and cluster identities were determined using the Seurat FindAllMarkers function using the Wilcoxon rank sum test defining cluster marker genes as genes expressed in at least 25% of the cells of either of the two compared populations, with a difference of at least 10% between the populations, a log fold change > 0.25. During log fold change calculation, the mean was computed as described above. Subsequently, cells were annotated in a two-step process based on cluster marker genes and protein marker expression as CD4 naïve (*CCR7*, *LEF1*, CD4, CD45RA, CCR7, CD62L), CD4 memory (*AQP3*, CD4, CD25), Activated CD4 memory (*AQP3*, *CXCR3*, *ANXA5*, CD4, CD28, HLA-DR), *GZMH*⁺ CD4 memory (*GZMH*, *FGFBP2*, CD4), Treg (*FOXP3*, *IL2RA*, *IKZF2*, CD4, CD25), CD8 naïve (*CD8B*, *CCR7*, *LEF1*, CD8, CD45RA, CCR7, CD62L), *GZMK*⁺ CD8 memory (*CD8B*, *GZMK*, CD8), *GZMH*⁺ CD8 memory (*CD8B*, *GZMH*, *FGFBP2*, CD8), MAIT (*KLRB1*, *SLC4A10*, CD8, CD161), gamma-delta (gd) (*TRGC1*, *TRGC2*, *TRDC*, CD3), CD4-CD8-double-negative (dn) (CD3, CD27, CD28), CD56^dim NK (*KLRF1*, *IL2RB*, CD56, CD16), CD56^bright NK (*KLRF1*, *XCL2*, *NCAM1*, CD56).

For differential expression analysis, cluster annotations of memory T cells were summarized into broader categories to ensure sufficient cell numbers per compared group and cluster (≥100 cells), as follows: CD4 memory (CD4 memory, Activated CD4 memory, *GZMH*⁺ CD4 memory), CD8 memory (*GZMK*⁺ CD8 memory, *GZMH*⁺CD8 memory).

## Annotation and integration of other disease scRNA-seq datasets

The datasets from Schulte-Schrepping et al. (COVID-19, EGAS0000 1004571)[32], Zhang et al. (Influenza, GSE243629)[78], and Yanagihara et al. (lung cancer, GSE215219)[79] were processed based on the steps used for processing the T and NK cell subset of our scRNA-seq EPTB dataset. Based on the respective dataset's distribution, the lung cancer dataset's cells were additionally filtered for > 250 genes and 10 % mitochondrial counts, while the Influenza dataset's cells were filtered for > 250 genes, 20 % mitochondrial and 10 % hemoglobin counts. For both the lung cancer and Influenza datasets, the harmony[80] (v0.1.0) algorithm was applied in downstream processing to account for donor-specific batch effects. All datasets were subset for T and NK cells based on literature-known marker gene expression in an iterative clustering and filtering approach, as described before and using the following parameters: COVID-19 dataset: first 10 PCs, resolution 1.2;

Influenza dataset: first 20 corrected PCs, resolution 0.4, then first 13 corrected PCs, resolution 0.7, then first 12 corrected PCs, resolution 0.7; lung cancer dataset: first 15 PCs, resolution 0.7, then first 13 corrected PCs, resolution 0.3. Final cell type annotation was performed separately for each dataset using Seurat reference mapping (Find-TransferAnchors and MapQuery functions) from the EPTB dataset based on the first 20 PCs. Integration of all datasets was done for a combined UMAP visualization based on the first 10 PCs using reciprocal PCA. The final dataset contained 63,019 cells from the COVID-19 dataset (n = 13 controls, 8 mild COVID, 9 severe COVID patients), 22,670 cells from the Influenza dataset (n = 2 controls, 2 pregnant controls, 2 Influenza-infected children, 5 Influenza-infected adults), 64,559 cells from the lung cancer dataset (n = 4 patients) and the 29,662 cells from the EPTB dataset. Pregnant controls were not included in subsequent analyses. One control and a mild COVID donor each were excluded from cell state quantification due to a cell number < 100.

## Correlation analysis of cell type frequencies
Correlation between measured (flow data and scRNA-seq) and estimated (bulk RNA-seq) cell type frequencies of patients present in all data modalities was calculated using Pearson's correlation coefficient *r*.

## Differential expression analysis in scRNA-seq data
Differential expression analysis was performed based on the Seurat FindMarkers with a Wilcoxon rank sum test. Differentially expressed genes were defined as genes with an abs(log2 fold change) > 0.25, were expressed in > 10 % of cells in the group of interest, and were below the 0.05 threshold for the bonferroni-adjusted *p*-values. Overrepresentation of DEGs in functional terms of the GO data base[52,53], KEGG[54–56] data base and Molecular Signature Database Hallmark gene sets [57,58] was assessed using the clusterProfiler[59] (v4.0.5) *enricher* function. Terms with a bonferroni-adjusted *p*-value < 0.05 were set to be significant.

## Module score calculation
Module scores were calculated using the R/Seurat AddModuleScore function at default settings for each respective signature. For the enrichment of the C1Q$^+$ intermediate monocyte signature, the top 20 upregulated markers from Hillman et al.[35] were enriched per monocyte state. Statistics were computed using a Kruskal-Wallis test on donor level.

## Pathway activity analysis
TNFα pathway activity was inferred using the PROGENy[81] (v1.22.0) and decoupleR[82] (v2.6.0) R packages in R (v4.3.0) and R Studio (v2023.3.0.386). Based on the PROGENy curated collection of pathways and target genes (n = 500 per pathway), a multivariate linear model was fit for each cell using the normalized gene expression data matrix as input. The model predicted gene expression based on pathway-gene interaction weights, obtaining t-values that were interpreted as pathway activity scores.

## Luminex cytokine array
Luminex Discovery Assay (R&D Biotechne, Minneapolis, MN, USA) was performed according the manufacturer's instructions. Samples were thawed on ice and centrifuged at 1000 × *g* for 5 min. Prior to analysis, samples were diluted 1:4 with Calibrator Dilutent (R&D Biotechne). All analytes were measured with Luminex 200 xMAP system (Thermo Fisher Scientific, Waltham, MA, USA) in technical duplicates. Standard curves were performed for all analytes, and xPONENT software (Thermo Fisher Scientific) was used for data collection and analysis. GraphPad Prism 9.5.1 (GraphPad) was used to analyse the data.

## Flow cytometry and cluster analysis
PBMCs were thawed and washed (centrifugation at 400 x *g* for 5 min, for all washing steps) with DPBS. Next, PBMCs were blocked 20 min on ice with DPBS containing 10% FBS and 5 mM EDTA. PBMCs were washed with DPBS containing 1% FBS and 5 mM EDTA and stained with previously titrated antibodies (all antibodies are listed in Supplementary Table 3) in DPBS containing 1% FBS and 5 mM EDTA for 30 min at 4 °C. Afterwards, cells were washed and directly acquired using a S3 Symphony Cytometer (BD, Heidelberg, Germany). Cytometer settings were kept the same for all experiments, and data was analysed using FlowJo software (BD) (Version 10.8.1).

To verify identified cell types and visualize changes, unsupervised clustering was carried out. For this, the FlowSOM clustering algorithm[83] (version 4.0.0, available at https://www.flowjo.com/exchange/#/plugin/profile?id=7) was applied to the cells based on a chosen set of lineage-defining markers for each cell type. The cells were clustered into 100 different groups, which were visualized in a minimal spanning tree. From these, a biologically appropriate number of metaclusters for the respective cell types was generated. Metaclusters were named based on the included cell's relative expression of defining surface markers. To circumvent bias caused by the variable number of patients within the different groups, the number of included cells was equalized before clustering using the FlowJo DownSample plugin (Version 3.3.1, available at https://www.flowjo.com/exchange/#/plugin/profile?id=25). In order to better visualize group-specific changes in individual clusters, the cell data was reduced to two dimensions using the t-distributed stochastic neighbor embedding (t-SNE) algorithm in FlowJo[84]. The acquired FlowSOM metaclusters were then projected onto the respective t-SNE blots.

## Statistical analysis
Statistical analysis was performed using GraphPad Prism 9.5.1 software (GraphPad) and the R package rstatix[85] (v0.7.0). For all experiments, full statistical parameters (value of n, standard deviation and statistical tests) are provided within the corresponding figure legends. For the whole study, *p*-values less than or equal to 0.05 were considered statistically significant and numerical *p*-values are shown within the figures. All data points represent biologically independent data points. Graphical illustrations were created with BioRender.com.

## Ethics
The study is part of the Cologne EX-TB project (ClinicalTrials.gov Identifier: NCT06875336). All procedures involving human participants were conducted in accordance with the ethical standards of the institutional research committee, the 1964 Declaration of Helsinki and its later amendments, and the principles outlined in the Belmont Report. Written informed consent was obtained from all study participants prior to inclusion. Individuals gave consent that pseudonymised data can be provided in the manuscript. The study protocol, methodology, and all associated documents (e.g., informed consent form) were approved by the ethics committee of the University Hospital Cologne (identifier: 18-079).

## Reporting summary
Further information on research design is available in the Nature Portfolio Reporting Summary linked to this article.

# Data availability
For the integration of scRNA-seq datasets, the following additional publically available datasets were used: EGAS00001004571, GSE243629, and GSE215219. For the integration of 'Other Disease' datasets, the following additional publically available datasets were used: GSE42834, GSE68310, EGAS00001004503, and GSE154918. The whole blood RNA-seq and scRNA-seq data generated in this study have

been deposited at the European Genome-phenome Archive (EGA) under accession codes EGAS50000000668 and EGAS50000000758, respectively, which is hosted by the EBI and the CRG. Restrictions apply regarding the availability of patient-derived biosamples. Source data are provided with this paper. Requests for access to biological material should be directed to the corresponding authors. Requests are subject to ethical review and approval by the Ethics Committee of the University Hospital Cologne, and certain restrictions may apply. Source data are provided in this paper.

## Code availability

All original code has been deposited at GitLab under https://gitlab. dzne.de/ag-ulas/EPTB-RNAseq-Analysis and Zenodo under https://doi. org/10.5281/zenodo.17240509.

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

## Acknowledgements

We thank all participants of this study and the study nurses. We acknowledge Melanie Nuesch-Gamaro for her help with data inspection. We are grateful to Prof. Anne O'Garra (The Francis Crick Institute) for her

critical reading of the manuscript. The author(s) declare financial support was received for the research and publication of this article. The author's research is funded by the European Union Innovative Medicines Initiative 2 Joint Undertaking program grant no. 853989 (ERA4TB to J.R.), the German Federal Ministry of Education and Research (BMBF; grant IdEpiCo to J.R.), the Deutsche Forschungsgemeinscchaft (DFG, German Research Foundation, CRC1403 project number 414786233 to J.R.), and the German Center for Infection Research (DZIF; TTU 02.814 and 02.913 to J.R.). S.T. and J.R. are supported by a research grant of the CMMC (B10), and S.T. by stipends from the Imhoff-Stiftung, Koeln Fortune Program and Jubiläumstiftung 1988.

## Author contributions

Conceptualization: J.R. and I.S.; Methodology: S.J.T., K.D., D.L., E.D.D., H.W., M.v.U., J.L.S., M.D.B., T.U., and J.R., Investigation: S.J.T., K.D., D.L., J.B.S., S.W., A.K., L.H., A.S., T.U., I.S., and J.R.; Visualization: S.J.T., K.D., D.L., J.B.S., T.U., and J.R.; Funding acquisition: I.S. and J.R.; Project administration: T.U., I.S., and J.R.; Supervision: T.U. and J.R.; Writing-original draft: S.J.T., K.D., D.L., J.B.S., L.H., T.U., and J.R.; Writing- review & editing: S.J.T., K.D., D.L., J.B.S., L.H., J.L.S., T.U., J.R. S.J.T., K.D., D.L., I.S., and J.R. contributed equally to this work.

## Funding

## Competing interests

The authors declare that the research was conducted in the absence of any commercial or financial relationships that could be construed as a potential competing of interest. Certain data and concepts presented here are part of a patent-pending application.
