## [Transparent Peer Review file · Nature Communications]

Deep immune profiling delineates hallmarks of disease heterogeneity in extrapulmonary tuberculosis

Corresponding Author: Professor Jan Rybniker

Version 0:

Reviewer comments:

Reviewer #1

(Remarks to the Author)
Strength of the Article:

Overall, the article is well-structured and provides in-depth information on EPTB and its infectious spectrum. The discussion of transcriptomics and immunological findings offers crucial insights into how various immune cells function in synchrony, as well as how their loss of coordination contributes to disease pathogenesis.

The authors effectively handle the data, giving equal weight to each component of their analysis while providing well-reasoned justifications for their interpretations. The integration of sequencing techniques (bulk and scRNA-seq), flow cytometry, and multiplex cytokine assays is a significant strength of the study, allowing for a deeper understanding of the nuanced differences within this heterogeneous disease. The staging of disease using gene expression data, along with the EPTB core signature that specifically differentiates EPTB and PTB from other diseases, provides valuable clinical insights. These findings can aid in more precise diagnostics and guide the development of targeted and effective treatment strategies.

Weakness of the Article:

- 1) Why were only upregulated genes emphasized, while downregulated genes and their associated pathways were not considered in the results or discussion?
Downregulated genes play an equally important role in the disease setting, as they can reveal critical disruptions in cellular functions, regulatory pathways, and immune responses. Completely overlooking their contribution presents a biased approach, potentially leading to an incomplete understanding of disease mechanisms.
- 2) Since the study includes scRNA-seq analysis, incorporating cell-to-cell communication analysis would have provided deeper insights into intercellular interactions, signaling pathways, and immune crosstalk. This approach could have enhanced the understanding of how different cell types coordinate or fail to do so in the disease context, offering a more comprehensive perspective on disease pathogenesis.

Major errors:

Line no.613: Fold change > 2 is mentioned and how about fold change <2?

Line no.821: Similar to previous comment, log2fold change > 0.25 mentioned and how about <0.25?

Minor errors in the article:

Line no. 205 : Figure 2e must be 2d

Line no. 211: EPTB spelling and also check in the overall document as well

Line 1092-1094: All three volcano plots (g, h and i) mentioned as mild

(Remarks on code availability)

Not aware of this code

Reviewer #2

(Remarks to the Author)

This manuscript describes extensive immune-phenotyping study of 29 adults with extra-pulmonary TB. Whole blood RNA seq and PBMC single cell sequencing were performed on all participants, providing an extremely rich dataset. The manuscript contains much descriptive immunological phenotyping, of both transcriptional and cellular response. I lack the expertise to review the immunology in detail. However, underpinning the data, and the analyses conducted, is the assertion that these 29 patients can be meaningfully divided into three categories of disease severity: mild, intermediate and severe. In so doing, the authors suggest there are disease sub-types that may require different therapeutic approaches. The challenges I have with this approach are as follows:

1. It is unclear to me how severity was assessed – there is very little or no information in the methods. What information was used and were the criteria defined a priori? There is a suggestion in the results that the criteria were not just clinical, but included the inflammatory phenotyping data itself (line 143-145). If this is the case, then what weight was given to the clinical features and how did they influence the severity assessment? There is a danger of circular logic here. The clinical phenotyping is as essential as the immunological phenotyping to the analysis and conclusions, but at present it is poorly described.

2. The investigators have combining isolated cervical lymph node disease (N=12) with bone and joint, pleural, abdominal, and pericardial TB and (seemingly) rather arbitrarily defined the cases as mild, moderate and severe. Does the severity classification make any clinical or biological sense? There are rather scant and difficult to digest information on the 29 participants (an excel file in the supplement – I suggest this is converted to a summary table). TB, especially EPTB, is difficult to phenotype clinically, as the extent of the disease (organ involvement) can be hard to determine unless there is systematic and rigorous whole-body imaging. Furthermore, disease severity does not necessarily correlate with organs involved. The best example of this is brain TB: a single organ is involved in many cases, but death occurs in around 30%: it is a severe disease, even in its mildest form.

Nevertheless, if the investigators used disease extent as an indicator of severity, each patient should get the same investigations. I am uncertain from the information provided in the methods section whether this was done. But the question remains as to whether it is clinically and biologically meaningful to define severity on the numbers of different organs/lymph nodes involved.

3. A further limitation of the analyses presented, and the conclusions drawn, is that they arise from a single cohort. None of the findings are validated in a second cohort, although there is an attempt to provide some validation of 'diagnostic' transcriptional signatures in previously published datasets. The findings arise from a relatively small number of patients, without external validation, and are therefore preliminary in nature.

(Remarks on code availability)

Reviewer #3

(Remarks to the Author)

Summary

Theobald et al. set out to characterize blood immune signatures of EPTB using single-cell profiling technologies. The topic is of high interest as the vast majority of blood profiling studies in TB are only available for PTB. Authors divided EPTB in three groups based on bulk transcriptomic profiles and associated them with clinical severity. Authors then performed single-cell RNA and protein analyses to identify gene signatures, cell subsets and pathways associated with EPTB severity groups. Authors generated a significant amount of data, which is unfortunately not sufficiently integrated together. In many sections, the writing lack clarity about which comparisons are being referred to and is often difficult to follow (especially with so much data to show). There are several claims not supported by the data. Importantly, a key comparison with PTB throughout is missing to ascertain that all immune signatures reported here are EPTB specific and not general TB. Diagnostic signatures show similar performance to previously published signatures of PTB so the significance is low. There is also a lack of data demonstrating that the EPTB groups indeed reflect clinical severity.

Major comments:

1. I have a problem with the definition of severity in EPTB: it seems quite a bold claim solely based on the # of organs affected. Does multiple organs really indicate more severity? Are there any other clinical data to support this claim? In addition, the breakdown into groups was done based on bulk transcriptomic profiles, not clinical data. A more accurate definition would be to call the groups "inflammatory phenotype" low/int/high rather than "disease severity" mild/int/severe (which implies the use of clinical parameters). Alternatively, authors could divide the patients into severity groups based on the clinical data and then show that this grouping associates with distinct transcriptomic profiles.

2. How do the authors reconcile that at several instances they find differences more significant in intermediate compared to mild or severe groups, but then they describe that changes are "progressive" across groups? Also, the wording "progressive" or "continuum" implies that there is a temporal switch between EPTB "severity" states, which I am not sure is clinically true (especially based on the authors definition of severity, see point 1), and definitely not proven here (this is a cross-sectional study).

3. The flow cytometry data analysis adds little – and its presentation is confusing. What do the intricate flow cytometry analyses presented add to the single-cell RNAseq data analyses, which allow a much higher resolution? High-dimensional analyses need to be more detailed (2D projection of clusters, justification for cluster annotation, how the frequency of each cluster change between groups etc...similar to single-cell RNAseq analysis workflows). The FlowSOM and tSNE plots are not interpretable presented as is. The flow cytometry gating strategy also lack clarity (some arrows and plots are missing).

4. Importantly, how does the single-cell flow and RNA data correlate with the bulk RNAseq data? Authors need to integrate

the different data types as my understanding is that a subset of the patients analyzed in bulk were used for the single-cell analyses.

5. Importantly, authors need to show individual patient variation for all analyses (as Fig 1 clearly show that there is a high inter-individual variability even within groups). This is essential for cell cluster composition to ensure that the enrichment for certain cell types is contributed by all (or the majority) of the patients per group.

6. More information on cluster annotation is needed (for both flow cytometry and RNA analyses)

7. Some clusters/cell populations appear to be cherry-picked for further analysis without logic/strong rationale. (e.g., focus on GZMH CD4 T cells; or grouping into naïve/memory T cell groups for Fig 5c-d-f while the rest of the data presented in Fig 5 is per cluster)

8. Several claims are not supported by the data. Including but not limited to: lines 275-277: no figure is comparing B cell cluster frequencies across groups; Fig 5e or 5f do not show association between GZMH CD4 T cells and severity groups as claimed in lines 356-360; Fig 5h or Ext data Fig 6f do not show that the frequency of the activated CD4 cells in the meta-analysis is indeed lower than in the authors' dataset so the statement line 373 cannot be verified; Fig 5f does not show a gradual increase with severity as claimed in line 359, etc...

9. The single-cell analyses are missing a key comparison with PTB. In addition, the diagnostic signature of EPTB does not show discriminatory power between EPTB vs PTB. So how can the authors ascertain that what they are looking at is EPTB specific (as claimed) and not general TB signatures? Blood signatures associated with IFN signaling, activated myeloid cells, activated T cells...have all been widely described for PTB. Also, the diagnostic signature shows similar performance to previously published PTB signatures, so its significance is low.

(Remarks on code availability)

N/A

Version 1:

Reviewer comments:

Reviewer #1

(Remarks to the Author)

The authors have incorporated the suggested changes regarding down regulated genes and also included the additional results they obtained in the respective sections.

In the case of cell-cell communication analysis, the authors claim they have obtained the same information similar to that of the sequencing results and couldn't obtain any additional information. With reference to the authors response, perhaps, that part can be avoided in incorporating to the manuscript.

However, some minor corrections are still pending in the revised document.

- The fold change < 2. or log fold change < 0.25 value for downregulation has not been mentioned in the methodology section.
 - The figure 1i to 1k is wrongly mentioned as INF low category for all the three volcano plots. Kindly rectify with low, intermediate and high category.
 - Extended data Table 2 and 3 is not available in the supporting document or mistakenly named as Table 4 and with their further continuation.
 - The sentence in the abstract is a repetition. Check and remove it.
- "Combining bulk with single-cell RNA-sequencing delineated immunological trajectories which were characterized by dynamic IFN and IL-1-mediated signalling in monocytes in conjunction with hyperactivation of T and NK cells eventually resulting in extensive immune-dysregulation.

(Remarks on code availability)

NA

Reviewer #2

(Remarks to the Author)

The authors have addressed my concerns. The move away from defining disease severity to immunotypes makes sense, and improves the coherence of the manuscript.

(Remarks on code availability)

Reviewer #3

(Remarks to the Author)

Thank you to the authors for addressing my comments. I appreciate the time and effort taken to rename the classification groups, include additional details on the flow cytometry data analysis, and clarify the focus on specific cell types.

My remaining concerns pertain to the flow cytometry data, which is still not presented objectively. This is important if the dataset is to serve as a valuable resource for other scientists working with similar patient-derived samples, as suggested in the rebuttal letter.

1. Figs 3b, S6c and S6d show 10 monocyte clusters, including 8 with an "annotation", but why only 5 were considered biologically relevant in the main text? Why some were annotated, some not? CD14+CD163+DR+ (annotation for the light green cluster) was also the annotation for cluster 3, so why ignoring this cluster? I could not find anywhere information on how and why this decision was made. In addition, in the main text, authors states lines 253-254 that only 5 clusters of monocytes were identified which is misleading and should be rephrased. I have the same questions for B cell clusters shown in Fig S7.
2. The FlowSOM spanning trees shown in Fig 3b, and in Figs S6-S7 do not effectively illustrate marker expression within clusters or differences between groups. For marker expression in clusters, use heatmaps instead, as done in Fig S7a for T cells, or dot plots as in Fig S8a. For cluster composition differences between groups, stick to frequency graphs as shown in Fig S8d, showing individual patient data. Consider removing the tree plots altogether, they take up a lot of space but are difficult to interpret (and not legible at the current resolution).
3. Figure S7a refers to 16 clusters from a tSNE analysis in Fig 3. There is no panel in Fig 3 that is showing a tSNE plot depicting the 16 T cell clusters? In addition, please show the composition differences across groups for all 16 clusters, not only selected clusters 1, 11 and 12.
4. The cluster composition graphs in Fig 3a, Fig S6bf and S7cfj should show individual datapoints for each patient.

Other concerns

5. Biological sex information is not available in Table S1 as indicated in methods lines 609-610.
6. Correlation analysis lines 314-316: indicate which datasets were compared (i.e., single-cell RNAseq compared to flow cytometry and bulk RNAseq)
7. It is not clear why a PTB cohort was not included for comparison in Fig 5i/ Fig S12. It should be at least mentioned in the discussion that whether the aCD4m T cell subset is EPTB specific, or general TB-specific, remain to be determined.
8. Lines 533-534: this is a stretch given the current study design. The standard workflow for identifying diagnostic signatures is to have a much larger cohort, and divide it into training, test and validation sets. This should be the next step. Please indicate that further validation of the diagnostic signature is warranted before moving into clinical trials. Include in the limitations of the study as well.
9. The figures are at too low resolution and the font is too small. It is not always possible to decipher the + from the -, and some of the writing.
10. Line 271, replace lymphocyte with T cell? NK cells (discussed in the previous paragraph) are also lymphocytes.
11. Typo line 333: should be Fig 10d?
12. Lines 513-520, missing reference.

(Remarks on code availability)

The link for the code is invalid.

Version 2:

Reviewer comments:

Reviewer #3

(Remarks to the Author)

(Remarks on code availability)

All my comments have been addressed.

Rebuttal for Manuscript reference number: NCOMMS-25-13033-T

Title: Deep immune profiling delineates hallmarks of disease heterogeneity in extrapulmonary tuberculosis

REVIEWER COMMENTS

Reviewer #1 (Remarks to the Author):

Strength of the Article:

Overall, the article is well-structured and provides in-depth information on EPTB and its infectious spectrum. The discussion of transcriptomics and immunological findings offers crucial insights into how various immune cells function in synchrony, as well as how their loss of coordination contributes to disease pathogenesis.

The authors effectively handle the data, giving equal weight to each component of their analysis while providing well-reasoned justifications for their interpretations. The integration of sequencing techniques (bulk and scRNA-seq), flow cytometry, and multiplex cytokine assays is a significant strength of the study, allowing for a deeper understanding of the nuanced differences within this heterogeneous disease. The staging of disease using gene expression data, along with the EPTB core signature that specifically differentiates EPTB and PTB from other diseases, provides valuable clinical insights. These findings can aid in more precise diagnostics and guide the development of targeted and effective treatment strategies.

Weakness of the Article:

1) Why were only upregulated genes emphasized, while downregulated genes and their associated pathways were not considered in the results or discussion?

Downregulated genes play an equally important role in the disease setting, as they can reveal critical disruptions in cellular functions, regulatory pathways, and immune responses. Completely overlooking their contribution presents a biased approach, potentially leading to an incomplete understanding of disease mechanisms.

Our response:

We are thankful for the comments of the reviewer and agree with them on the importance of downregulated genes in disease settings. Therefore, we added additional panels assessing the biological processes and functions associated with the downregulated genes of mild, intermediate, and severe EPTB (from hereon called low “INF_{low}”, intermediate “INF_{int}”, and high “INF_{high}” inflammatory immunotype, respectively, to avoid misinterpretation of our data, see comments for reviewer 2 and 3). Inspecting the intersection of the downregulated genes shared between INF_{int} and INF_{high} EPTB in our bulk RNA-seq data additionally highlighted the downregulation of B cell activity (new **Extended Data Fig. 1c**, in lines 188-190) in line with the reduced frequency of B cells observed in the FACS and scRNA-seq data (**Fig. 3a**, new **Extended Data Fig. 8d**) as well as estimated B cell frequencies from the cell type deconvolution (**Fig. 2e**). In addition, enrichment of the downregulated genes identified in major cell types of the scRNA-seq data (cell types with more than 2,000 cells) revealed a significant association to TNF α signaling via NF κ B (new **Extended Data Fig. 8b**). To further substantiate

this finding, we inferred the overall activity of the TNF α signaling pathway using a signature footprint-based approach which leverages gene expression profiles and a core of pathway-responsive gene signatures instead of immediate downstream elements (**Extended Data Fig. 8c**)¹. This is now described in lines 307-310. While a relatively reduced pathway activity was observed in CD4 and CD8 T cells, NK cells, and B cells, an increased TNF α pathway activity was measured in CD14 monocytes. Similarly, the downregulated DEGs between the EPTB immunotypes and healthy controls within the CD14⁺IF1⁺ monocytes were also associated with TNF α signaling via NF κ B (Response Letter **Figure 1a**, new **Extended Data Table 4**), yet, the footprint-based approach depicted that, in the majority of these cells, no change in the pathway activity was measured (Response Letter **Figure 1b**). Furthermore, this enrichment revealed the downregulation of genes associated with the oxidative phosphorylation across inflammatory stages (Response Letter **Figure 1a**, new **Extended Data Table 4**) pointing towards a metabolic switch in EPTB patients.

Figure 1 Downregulated genes point towards metabolic switch in CD14⁺IF1⁺ monocytes

(A) Dot plot of functional enrichment of downregulated genes in CD14⁺IF1⁺ monocytes between EPTB inflammatory stages and healthy controls using the GO biological processes, KEGG and Molecular Signature Database Hallmark gene set databases. The top 10 terms that were significantly enriched per comparison are displayed ordered by adjusted p-value (Benjamini-Hochberg corrected p-value <0.05). B) Density plot of TNF α pathway activity scores across CD14⁺IF1⁺ monocytes. The red line represents the median pathway activity score.

2) Since the study includes scRNA-seq analysis, incorporating cell-to-cell communication analysis would have provided deeper insights into intercellular interactions, signaling pathways, and immune crosstalk. This approach could have enhanced the understanding of how different cell types coordinate or fail to do so in the disease context, offering a more comprehensive perspective on disease pathogenesis.

Our response:

We thank the reviewer for their suggestion and bringing up the topic of cell-cell communication. To follow up on the comment, we assessed cell-cell communication (CCC) within the healthy controls and EPTB samples of our scRNA-seq PBMC data using CellChat² (Response letter **Figure 2A+B**). Independent of the disease, CD14 and CD16 monocytes were identified as the major source of outgoing signals and receiver of incoming signals with an enhanced incoming interaction strength in EPTB further highlighting the role of monocytes in EPTB (Response letter **Figure 1B**). Similarly, an increase in outgoing signals was also observed in NK cells of EPTB patients compared to healthy

controls (Response letter **Figure 2B**) in line with upregulation of differentially expressed genes (DEGs) of NK cells involved in T cell activation and interferon (IFN) response in the intermediate and high inflammation immunotype (**Fig. 5d and 5e, Extended Data Fig. 11e**). In contrast, a prominent loss of the outgoing interaction strength in B cells was measured in EPTB (Response letter **Figure 2B**) matching the decreased frequencies of B cells observed in the intermediate and high inflammation immunotype (**Fig. 4e**).

Overall, the findings of the CCC analysis are in line with our results from bulk RNA-seq and sc RNA-seq. As the CCC analysis did not reveal novel insights, we leave the decision for the inclusion of the CCC analysis in the manuscript to the discretion of the editor.

Figure 2 Assessment of cell-cell communication reveals monocytes as key component

(A+B) Dot plot of the outgoing and incoming interaction strength of scRNA-seq PBMCs in healthy controls and EPTB patients, respectively, based on a cell-cell communication analysis. Dots are scaled by the number of interactions and colored based on the cell type. Only Cell types with more than 2,000 cells were selected for the analysis.

Major errors:

Line no.613: Fold change > 2 is mentioned and how about fold change <2?

Line no.821: Similar to previous comment, log2fold change > 0.25 mentioned and how about <0.25?

Our response:

We appreciate the reviewer for indicating this error and for noting the missing precision in the method section. To this end, we have adapted these sections to mention the absolute fold change > 2 and the absolute log2 fold change > 0.25 in lines 660 and 926, respectively. Figure legends were also corrected accordingly.

Minor errors in the article:

Line no. 205 : Figure 2e must be 2d

Line no. 211: EPTB spelling and also check in the overall document as well

Line 1092-1094: All three volcano plots (g, h and i) mentioned as mild

Our response:

We thank the reviewer for pointing out our mistake. We have corrected and adapted the manuscript at the respective positions accordingly.

Reviewer #1 (Remarks on code availability):

Not aware of this code

Our response:

We have now included an access token for the created GitHub repository (<https://github.com/schultzelab/EPTB-RNAseq-Analysis>) for further inspection of the code.

Reviewer #2 (Remarks to the Author):

This manuscript describes extensive immune-phenotyping study of 29 adults with extra-pulmonary TB. Whole blood RNA seq and PBMC single cell sequencing were performed on all participants, providing an extremely rich dataset. The manuscript contains much descriptive immunological phenotyping, of both transcriptional and cellular response. I lack the expertise to review the immunology in detail. However, underpinning the data, and the analyses conducted, is the assertion that these 29 patients can be meaningfully divided into three categories of disease severity: mild, intermediate and severe. In so doing, the authors suggest there are disease sub-types that may require different therapeutic approaches. The challenges I have with this approach are as follows:

1. It is unclear to me how severity was assessed – there is very little or no information in the methods. What information was used and were the criteria defined a priori? There is a suggestion in the results that the criteria were not just clinical, but included the inflammatory phenotyping data itself (line 143-145). If this is the case, then what weight was given to the clinical features and how did they influence the severity assessment? There is a danger of circular logic here. The clinical phenotyping is as essential as the immunological phenotyping to the analysis and conclusions, but at present it is poorly described.

Our response:

We thank reviewer 2 for this important point of critique. We apologize for not having described this sufficiently clear. It is important to mention that our prospective cohort includes clinically well characterized patients whose clinical data and blood values were collected by MDs of the study group using electronic case report files. Each patient underwent standardized imaging protocols at baseline which included at least a scan of the chest and abdominal ultrasound. Additional imaging was performed following rigorous assessment of symptoms and included MRI or CT-scans (e.g. of affected bones). This is now described in more detail in the methods section.

Nevertheless, for this study, we decided to use blood transcriptional data as a **starting point** for categorizing immunological features of patients. In our view, this approach worked extremely well with a sharp and significant separation of three highly distinct immunotypes. Only then, we correlated these groups to clinical data. Interestingly, the immunotype with the least inflammation based on RNA-seq contained primarily patients with cervical lymph node TB which is a highly localized and mild form of the disease. Patients showing the most pronounced inflammatory immunotype presented with disseminated TB. The group in between represented primarily patients with lymph node tuberculosis affecting several body sites, however, there was also overlap with group 1 and group 3 regarding affected body sites. Thus, clinical imaging-based grouping aligned to some extent with blood transcriptome-based grouping. However, the correlation was not at 100%, which is expected given the difficulties in imaging-based staging of this highly heterogeneous disease.

To fully address the points of critique made by reviewer 2, we did the following changes:

First, we correlated the three transcriptome-derived immunotypes with plasma C-reactive protein (CRP) values, a well-established inflammatory marker which is routinely determined in TB patients. CRP discriminated group 1 (low inflammation) and groups 2 and 3 significantly (novel **Fig. 1g**) and modified **Extended Data Table 1**. To also differentiate between groups 2 and 3, we were able to use a more specific blood-based inflammatory parameter: interleukin 1 β (IL-1 β). This cytokine is strongly associated with Mtb pathogenicity^{3,4}. Interestingly, only group 3 patients had elevated IL-1 β levels in their plasma (novel **Fig. 1g**). This finding correlates extremely well with the inflammatory trajectories we identified in these patients which is driven by upregulation of NLRP3 inflammasome associated genes. The NLRP3 inflammasome is required for IL-1 β activation and secretion.

Second, we adjusted the wording in the entire manuscript. To avoid misinterpretation of our data, we now use the expression “immunotypes” with three grades of inflammation (low “INF_{low}”, intermediate “INF_{int}”, high “INF_{high}”) instead of describing the groups as mild, intermediate and severe.

Finally, we included a graphical summary better explaining our approach (novel **Fig. 7**).

2. The investigators have combining isolated cervical lymph node disease (N=12) with bone and joint, pleural, abdominal, and pericardial TB and (seemingly) rather arbitrarily defined the cases as mild, moderate and severe. Does the severity classification make any clinical or biological sense? There are rather scant and difficult to digest information on the 29 participants (an excel file in the supplement – I suggest this is converted to a summary table). TB, especially EPTB, is difficult to phenotype clinically, as the extent of the disease (organ involvement) can be hard to determine unless there is systematic and rigorous whole-body imaging. Furthermore, disease severity does not necessarily correlate with organs involved. The best example of this is brain TB: a single organ is involved in many cases, but death occurs in around 30%: it is a severe disease, even in its mildest form.

Nevertheless, if the investigators used disease extent as an indicator of severity, each patient should get the same investigations. I am uncertain from the information provided in the methods section whether this was done. But the question remains as to whether it is clinically and biologically meaningful to define severity on the numbers of different organs/lymph nodes involved.

Our response:

We thank reviewer 2 for this comment. We would also like to refer to our response for the comment above.

Of note, we agree that CNS tuberculosis is a unique and special form of EPTB which interacts with the immune system of the brain and blood compartments may not fully mirror immunopathology of this disease. For this reason, we excluded patients with CNS TB from this study. This fact is now highlighted in the discussion where we included a limitations section.

However, our use of the term “severe” was not intended to reflect clinical severity in terms of mortality or morbidity risk. Rather, we used it to indicate the extent of dissemination, defined as the involvement of two or more non-contiguous anatomical sites, in line with established definitions in the literature (Wang et al., 2007; Suárez et al., Lancet Infect Dis 2019). This form of EPTB is considered a marker of systemic disease and heightened inflammatory burden, even if it does not always correlate directly with clinical outcomes.

To avoid misinterpretation, we have now removed the term “severity” from the manuscript and instead describe the transcriptomic groups as immunotypes with low, intermediate, and high inflammatory activity (low “INF_{low}”, intermediate “INF_{int}”, high “INF_{high}”).

3. A further limitation of the analyses presented, and the conclusions drawn, is that they arise from a single cohort. None of the findings are validated in a second cohort, although there is an attempt to provide some validation of ‘diagnostic’ transcriptional signatures in previously published datasets. The findings arise from a relatively small number of patients, without external validation, and are therefore preliminary in nature.

Our response:

We thank the reviewer for pointing this out. To address this topic, we investigated the performance of the EPTB-core in the data provided by Blankley et al.⁵, another study evaluating blood transcriptomes in EPTB patients which were generated using microarray data (Response Letter **Figure 3**). The data was subset for control and EPTB samples prior to data normalization. Inspection of the transcriptional alterations between the 61 controls and 47 EPTB patients included in the final dataset revealed a prominent shift in transcriptional programming along PC1 and an increased heterogeneity of the controls compared to our dataset (Response Letter **Figure 3A**). Next, we enriched the EPTB-core signature in controls and EPTB patients (Response letter **Figure 3B**). Even though several controls displayed a positive enrichment of the EPTB-core, highly significant differences between EPTB patients and controls were observed. This was further corroborated by classifying the patients using a random forest classifier where the EPTB-core achieved an area under the curve (AUC) > 0.95 in both the training and test datasets, matching the performance in our dataset (Response letter **Figure 3C**).

It is important to mention that in Blankley et al., the researchers also tried to define a diagnostic signature that differentiates between EPTB and sarcoidosis, a key differential diagnosis of EPTB. However, this was not possible based on the microarray dataset of Blankley et al. In contrast, our diagnostic signature derived from state-of-the-art whole blood RNA-sequencing differentiated sarcoidosis patients from EPTB patients with high significance (**Fig. 6h**) again highlighting the quality of our data.

A PCA - Blankley et al., 2016

B GSVA of EPTB-core

C AUROC of EPTB-core

Figure 3 Validation of EPTB-core in whole blood microarray data of EPTB patients from Blankley et al.

(A) Principal component analysis (PCA) of the 1,000 most variable genes of the whole blood microarray data by Blankley et al. ⁵. Dots are colored based on the disease status and sample distributions along the first two principal components is displayed. (B) Boxplot of the gene set variation analysis (GSVA) enrichment scores from the EPTB-core signature split by disease status. Statistics were computed using a Wilcoxon test. (C) Area under the receiver operating characteristic (AUROC) of the EPTB-core in the Blankley et al. data. The area under the curve (AUC) is displayed for both training and test dataset.

Reviewer #3 (Remarks to the Author):

Summary

Theobald et al. set out to characterize blood immune signatures of EPTB using single-cell profiling technologies. The topic is of high interest as the vast majority of blood profiling studies in TB are only available for PTB. Authors divided EPTB in three groups based on bulk transcriptomic profiles and associated them with clinical severity. Authors then performed single-cell RNA and protein analyses to identify gene signatures, cell subsets and pathways associated with EPTB severity groups. Authors generated a significant amount of data, which is unfortunately not sufficiently integrated together. In many sections, the writing lack clarity about which comparisons are being referred to and is often difficult to follow (especially with so much data to show). There are several claims not supported by the data. Importantly, a key comparison with PTB throughout is missing to ascertain that all immune signatures reported here are EPTB specific and not general TB. Diagnostic signatures show similar performance to previously published signatures of PTB so the significance is low. There is also a lack of data demonstrating that the EPTB groups indeed reflect clinical severity.

Major comments:

1. I have a problem with the definition of severity in EPTB: it seems quite a bold claim solely based on the # of organs affected. Does multiple organs really indicate more severity? Are there any other clinical data to support this claim? In addition, the breakdown into groups was done based on bulk transcriptomic profiles, not clinical data. A more accurate definition would be to call the groups "inflammatory phenotype" low/int/high rather than "disease severity" mild/int/severe (which implies the use of clinical parameters). Alternatively, authors could divide the patients into severity groups

based on the clinical data and then show that this grouping associates with distinct transcriptomic profiles.

Our response:

We thank the reviewer for this important comment. We agree that defining “severity” solely based on the number of affected organs can be problematic, particularly in EPTB, where clinical severity does not always correlate with anatomical spread. This point of critique was also raised by reviewer 2 and we addressed this in both, the revised version of our manuscript and the rebuttal. We also appreciate the suggestion to rather define the groups we identified as “immunotypes” rather than using a more clinical severity score. This was adjusted accordingly.

We would first like to clarify that our prospective cohort includes clinically well characterized patients whose clinical data and blood values were collected by MDs of the study group using electronic case report files. Each patient underwent standardized imaging protocols at baseline which included at least a scan of the chest and abdominal ultrasound. Additional imaging was performed following rigorous assessment of symptoms and included MRI or CT-scans (e.g. of affected bones). This is now described in more detail in the methods section (lines 601-607).

Nevertheless, for this study, we decided to use blood transcriptional data as a **starting point** for categorizing immunological features of patients which worked extremely well with a sharp and significant separation of three highly distinct immunotypes. Only then, we correlated these groups to clinical data. Interestingly, the immunotype with the least inflammation based on RNA-seq contained primarily patients with cervical lymph node TB which is a highly localized and mild form of the disease. Patients showing the most pronounced inflammatory immunotype presented with disseminated TB. The group in between represented primarily patients with lymph node tuberculosis affecting several body sites, however, there was also overlap with group 1 and group 3 regarding affected body sites. Thus, clinical imaging-based grouping aligned to some extent with blood transcriptome-based grouping. However, the correlation was not at 100%, which is expected given the difficulties in imaging-based staging of this highly heterogeneous disease.

To address the points of critique made by reviewer 3, we did the following changes:

First, we correlated the three transcriptome-derived immunotypes with plasma C-reactive protein (CRP) values, a well-established inflammatory marker which is routinely determined in TB patients. CRP discriminated group 1 (low inflammation) and groups 2 and 3 significantly (novel **Fig. 1g** and modified **Extended Data Table 1**). To also differentiate between groups 2 and 3, we were able to use a more specific blood-based inflammatory parameter: interleukin 1 β (IL-1 β). This cytokine is strongly associated with Mtb pathogenicity^{3,4}. Interestingly, only group 3 patients had elevated IL-1 β levels in their plasma (novel **Fig. 1g**). This finding correlates extremely well with the inflammatory trajectories we identified in these patients which is driven by upregulation of NLRP3 inflammasome associated genes. The NLRP3 inflammasome is required for IL-1 β activation and secretion.

Second, we adjusted the wording in the entire manuscript as suggested by the reviewer. To avoid misinterpretation of our data, we now use the expression “immunotypes” with three grades of inflammation (low “INF_{low}”, intermediate “INF_{int}”, high “INF_{high}”) instead of describing the groups as mild, intermediate and severe.

2. How do the authors reconcile that at several instances they find differences more significant in intermediate compared to mild or severe groups, but then they describe that changes are “progressive” across groups? Also, the wording “progressive” or “continuum” implies that there is a temporal switch between EPTB “severity” states, which I am not sure is clinically true (especially based

on the authors definition of severity, see point 1), and definitely not proven here (this is a cross-sectional study).

Our response:

We agree that the terms “progressive” or “continuum” may imply a temporal evolution or clinical trajectory.

To avoid misinterpretation, we have revised the manuscript and now refer to the differences between groups as graded or stratified levels of inflammatory activation rather than using temporal wording like “progressive” or “continuum.”

3. The flow cytometry data analysis adds little – and its presentation is confusing. What do the intricate flow cytometry analyses presented add to the single-cell RNAseq data analyses, which allow a much higher resolution? High-dimensional analyses need to be more detailed (2D projection of clusters, justification for cluster annotation, how the frequency of each cluster change between groups etc...similar to single-cell RNAseq analysis workflows). The FlowSOM and tSNE plots are not interpretable presented as is. The flow cytometry gating strategy also lack clarity (some arrows and plots are missing).

Our response:

We thank the reviewer for raising this important point of critique. We extensively expanded our supplementary figures, now showing, as suggested by the reviewer, 2D projection of clusters, the relative expression of relevant markers within individual clusters, as well as graphs indicating how the frequencies of each cluster change between groups, for all analysis represented in the initially submitted manuscript (new **Extended Data Fig. 6&7**). We now also labeled the highlighted clusters in the displayed tSNE blots to allow easier identification, as well as updating the gating strategy now showing representative plots for every population, along with respective arrows.

We agree with the reviewer that the flow cytometry data largely confirm our bulk and scRNAseq data, however, we feel that this data-set is of great value since it supports exclusively transcriptomic data with phenotypic data. It is important to mention that receptors being upregulated on the RNA level were also seen to be upregulated on the protein level on the cell surface (e.g. PD-L1). This is also the case for the cytokine arrays we performed. Thus, we strongly believe that FACS data are highly supportive for the key findings of this study. In addition, the data can be used as a valuable resource for scientists looking into similar patient derived samples.

4. Importantly, how does the single-cell flow and RNA data correlate with the bulk RNAseq data? Authors need to integrate the different data types as my understanding is that a subset of the patients analyzed in bulk were used for the single-cell analyses.

Our response:

We thank the reviewer for pointing out this missing link. To this end, we correlated the frequencies of the matched samples across the different modalities (new **Extended Data Fig. 9**). Strong correlations across monocytes, NK cells, T cells, and B cells were observed between the scRNA-seq and FACS data, emphasizing their comparability (new **Extended Data Fig. 9A**). Similarly, strong correlations were also seen between the estimated bulk RNA-seq frequencies of monocytes and B cells and the two single-cell modalities (new **Extended Data Fig. 9B+C**). A significant correlation of NK cell frequencies was only

observed between scRNA-seq and bulkRNA-seq data (new **Extended Data Fig. 9B**), as the FACS data contained one outlier in the intermediate inflammation immunotype ((new **Extended Data Fig. 9C**). In contrast, the comparison of CD4 and CD8 T cells returned only weak, non-significant associations independent of the single-cell modality (new **Extended Data Fig. 9B+C**). While T cell subsets are generally difficult to separate based on RNA-seq data alone, phenotypic plasticity or disease-induced dysregulations can additionally impede the deconvolution performance of CIBERSORT^{6,7}. Further, the deconvolution algorithm requires a signature matrix which is constructed by identifying robust differentially expressed genes and is further filtered by a linear support vector regression (LSVR) during the deconvolution, and contains a matrix of marker genes of all cell types included in the reference. Based on the parameterization of the algorithm, this filtering approach might select genes expressed on two or more closely related cell types or genes part of signaling pathways heavily active in several cell types. This ‘spill over’ effect⁸ could also explain the differences identified between the deconvolution approach and scRNA-seq.

Overall, the major trends observed in the scRNA-seq and FACS data were conserved in bulkRNA-seq data allowing for the comparison of findings across modalities.

5. Importantly, authors need to show individual patient variation for all analyses (as Fig 1 clearly show that there is a high inter-individual variability even within groups). This is essential for cell cluster composition to ensure that the enrichment for certain cell types is contributed by all (or the majority) of the patients per group.

Our response:

We agree with the reviewer on the importance on individual patient variation in our single-cell data and exchanged all barplots depicting cell type frequencies with boxplots reporting on the sample distributions (new **Fig. 4e and 4g**, new **Fig. 5b**, new **Extended Data Fig. 8d**, new **Extended Data Fig. 10b**, new **Extended Data Fig. 11b**, new **Extended Data Fig. 12b**).

6. More information on cluster annotation is needed (for both flow cytometry and RNA analyses)

Our response:

We thank the reviewer for this remark and now provide more detailed insights into the subset annotation of the scRNA-seq data. First, we provide the expression values per cluster of a surface marker protein panel read out (AbSeq) for the respective scRNA-seq data subsets (new **Extended Data Table 3**) in addition to the cluster marker genes. Moreover, the gene and protein markers used for annotations are listed per cluster in the methods section for cluster annotation (lines 822-828, 851-855, and 877-89).

7. Some clusters/cell populations appear to be cherry-picked for further analysis without logic/strong rationale. (e.g., focus on GZMH CD4 T cells; or grouping into naïve/memory T cell groups for Fig 5c-d-f while the rest of the data presented in Fig 5 is per cluster)

Our response:

We thank reviewer 3 for bringing up this point and apologize for the lack of clarity in our line of argumentation. We now complemented the methods section (lines 886-890) and figure legend of Figure 5 with information for the reasoning behind cell grouping for the respective analyses. In brief, DE and gene signature enrichment analyses per EPTB group have been performed on a broader

annotation level to ensure sufficient cell numbers for reliable results (> 100 cells per compared group and cluster, > 2000 cells per cluster overall, **Fig. 5c, 5e**). Frequency analyses and deconvolution of signatures refer to fine grained cell states to unveil drivers behind observed broad trends (new **Fig. 5b, 5f**), which are in particular *GZMH+* CD8 memory T cells and activated CD4 memory T cells, as described in lines 384-386. Moreover, we provided information on the downregulated genes and associated functional enrichments that, in comparison, further underline the relevance of the described upregulated patterns in the respective cell types (lines 380-383, new **Extended Data Fig. 11d,f**).

Overall, we streamlined and focused Figure 5 and its associated results section to remove ambiguity of descriptions. In particular we now emphasize that the CD8 T cell memory compartment is the major driver of observed changes towards the high inflammation EPTB immunotype when referring to *GZMH+* memory T cells (lines 358-361, 384).

8. Several claims are not supported by the data. Including but not limited to: lines 275-277: no figure is comparing B cell cluster frequencies across groups; Fig 5e or 5f do not show association between *GZMH* CD4 T cells and severity groups as claimed in lines 356-360; Fig 5h or Ext data Fig 6f do not show that the frequency of the activated CD4 cells in the meta-analysis is indeed lower than in the authors' dataset so the statement line 373 cannot be verified; Fig 5f does not show a gradual increase with severity as claimed in line 359, etc...

Our response:

We thank the reviewer for bringing up this point and in line with the previous point, we adapted ambiguous text parts to describe the claims supported by the data more precisely. Figure 5 related changes now include the following: As mentioned above, observations in the high inflammatory EPTB state are primarily driven by *GZMH+* CD8 T cells and not as much by *GZMH+* CD4 T cells (**Fig. 5b, 5f, Extended Data Fig. 11b**). Additionally, we adapted our description of gene expression trends seen in the high inflammation stage instead of a gradual progression (Fig. 5e, lines 372-376).

Regarding the independent disease cohort analyses, we now specify that only infectious diseases show no similar enrichment of activated CD4 T cells compared to the high inflammation EPTB immunotype (**Extended Data Fig. 12b**, lines 393-396).

9. The single-cell analyses are missing a key comparison with PTB.

Our response:

We thank the reviewer for raising this interesting point. As of now, we have compared the C1Q intermediate monocyte population emerging in the high inflammatory immunotype with a similar intermediate monocyte population from Hilmann et al.⁹ present in PTB (**Extended Data Fig. 10c**). Additionally, we have discussed the findings from our scRNA-seq data with published findings on PTB in lines 516-523. While similar shifts in cell type and cell state frequencies were observed between the EPTB inflammatory stages and severity levels of PTB patients as defined by Wang et al.¹⁰, the comparison highlights their differences in molecular programming. Future studies aiming for the integrated assessment of EPTB and PTB in a confounder-aware setting are required to fully elucidate the differences in transcriptional programming of blood immune cells strategically, however, this is beyond the scope of the current manuscript.

10. In addition, the diagnostic signature of EPTB does not show discriminatory power between EPTB vs PTB. So how can the authors ascertain that what they are looking at is EPTB specific (as claimed) and not general TB signatures? Blood signatures associated with IFN signaling, activated myeloid cells, activated T cells...have all been widely described for PTB. Also, the diagnostic signature shows similar performance to previously published PTB signatures, so its significance is low.

Our response:

We thank Reviewer 3 for this important remark, which helped us to further clarify our approach regarding diagnostic signatures. Our objective was not to identify signatures that differentiate between EPTB and PTB—such a distinction holds limited clinical relevance. Instead, our aim was first to evaluate how previously published signatures (derived from PTB patients) perform in an exclusive EPTB patient cohort. This analysis revealed that the performance of existing signatures varied considerably. In particular, those signatures with a low number of target genes failed to adequately distinguish between EPTB and healthy controls. We then leveraged our highly heterogeneous cohort to identify a novel signature capable of performing equally well in both EPTB and PTB cases (Fig. 6.i and extended data 12 g). This effort led to the identification of a 15-gene signature that outperformed existing ones across all evaluated parameters. Importantly, the number of genes required for a signature has practical implications for clinical application. Technically, it is challenging to reliably quantify large gene sets using automated PCR or similar methods. Notably, the second-best performing signature tested on our EPTB cohort requires the quantification of 40 differentially expressed genes. Therefore, we believe that our 15-gene signature represents a meaningful advancement over existing alternatives.

Reviewer #3 (Remarks on code availability):

N/A

Our response:

We have now included an access token for the created GitHub repository (<https://github.com/schultzelab/EPTB-RNAseq-Analysis>) for further inspection of the code.

References

- 1 Schubert, M. *et al.* Perturbation-response genes reveal signaling footprints in cancer gene expression. *Nat Commun* **9**, 20 (2018). <https://doi.org:10.1038/s41467-017-02391-6>
- 2 Jin, S. *et al.* Inference and analysis of cell-cell communication using CellChat. *Nat Commun* **12**, 1088 (2021). <https://doi.org:10.1038/s41467-021-21246-9>
- 3 Silverio, D., Goncalves, R., Appelberg, R. & Saraiva, M. Advances on the Role and Applications of Interleukin-1 in Tuberculosis. *mBio* **12**, e0313421 (2021). <https://doi.org:10.1128/mBio.03134-21>
- 4 Mayer-Barber, K. D. *et al.* Caspase-1 independent IL-1beta production is critical for host resistance to mycobacterium tuberculosis and does not require TLR signaling in vivo. *J Immunol* **184**, 3326-3330 (2010). <https://doi.org:10.4049/jimmunol.0904189>
- 5 Blankley, S. *et al.* The Transcriptional Signature of Active Tuberculosis Reflects Symptom Status in Extra-Pulmonary and Pulmonary Tuberculosis. *PLoS One* **11**, e0162220 (2016). <https://doi.org:10.1371/journal.pone.0162220>
- 6 Newman, A. M. *et al.* Robust enumeration of cell subsets from tissue expression profiles. *Nat Methods* **12**, 453-457 (2015). <https://doi.org:10.1038/nmeth.3337>
- 7 Newman, A. M. *et al.* Determining cell type abundance and expression from bulk tissues with digital cytometry. *Nat Biotechnol* **37**, 773-782 (2019). <https://doi.org:10.1038/s41587-019-0114-2>
- 8 Aran, D., Hu, Z. & Butte, A. J. xCell: digitally portraying the tissue cellular heterogeneity landscape. *Genome Biol* **18**, 220 (2017). <https://doi.org:10.1186/s13059-017-1349-1>
- 9 Hillman, H. *et al.* Single-cell profiling reveals distinct subsets of CD14+ monocytes drive blood immune signatures of active tuberculosis. *Front Immunol* **13**, 1087010 (2022). <https://doi.org:10.3389/fimmu.2022.1087010>
- 10 Wang, Y. *et al.* Systemic immune dysregulation in severe tuberculosis patients revealed by a single-cell transcriptome atlas. *J Infect* **86**, 421-438 (2023). <https://doi.org:10.1016/j.jinf.2023.03.020>

Rebuttal for Manuscript reference number: NCOMMS-25-13033-T

Title: Deep immune profiling delineates hallmarks of disease heterogeneity in extrapulmonary tuberculosis

REVIEWER COMMENTS – second revision

Reviewer #1 (Remarks to the Author):

The authors have incorporated the suggested changes regarding down regulated genes and also included the additional results they obtained in the respective sections.

In the case of cell- cell communication analysis, the authors claim they have obtained the same information similar to that of the sequencing results and couldn't obtain any additional information. with reference to the authors response, perhaps, that part can be avoided in incorporating to the manuscript.

Our response:

We thank the reviewer again the time to evaluate the manuscript.

However, some minor corrections are still pending in the revised document.

- The fold change < 2. or log fold change < 0.25 value for downregulation has not been mentioned in the methodology section.

Our response:

We thank the reviewer for this important comment. To clarify, both upregulated (fold change > 2 or log2 fold change > 0.25) and downregulated genes (fold change < -2 or log2 fold change < -0.25) are summarized as genes with an absolute fold change greater than 2 or an absolute log2 fold change greater than 0.25. This description is provided in the Methods sections (lines 665 and 932).

- The figure 1i to 1k is wrongly mentioned as INF low category for all the three volcano plots. Kindly rectify with low, intermediate and high category.

Our response:

We changed the respective caption in figure 1.

- Extended data Table 2 and 3 is not available in the supporting document or mistakenly named as Table 4 and with their further continuation.

Our response:

We thank the reviewer for this comment. Tables 2 and 3 had been uploaded as excel files due to the large file size. The supporting document contained the legend only. These excel files should be made available via the reviewer interface of the journal.

- The sentence in the abstract is a repetition. Check and remove it.

“Combining bulk with single-cell RNA-sequencing delineated immunological trajectories which were characterized by dynamic IFN and IL-1-mediated signalling in monocytes in conjunction with hyperactivation of T and NK cells eventually resulting in extensive immune-dysregulation.

Our response:

We apologize for the duplication. The sentence is now deleted.

Reviewer #2 (Remarks to the Author):

The authors have addressed my concerns. The move away from defining disease severity to immunotypes makes sense, and improves the coherence of the manuscript.

Our response:

We thank the reviewer again for this positive comment and for taking the time to evaluate the manuscript.

Reviewer #3 (Remarks to the Author):

Thank you to the authors for addressing my comments. I appreciate the time and effort taken to rename the classification groups, include additional details on the flow cytometry data analysis, and clarify the focus on specific cell types.

Our response:

We thank the reviewer for the positive comment and for the time spent evaluating the manuscript.

“My remaining concerns pertain to the flow cytometry data, which is still not presented objectively. This is important if the dataset is to serve as a valuable resource for other scientists working with similar patient-derived samples, as suggested in the rebuttal letter.

1a. Figs 3b, S6c and S6d show 10 monocyte clusters, including 8 with an “annotation”, but why only 5 were considered biologically relevant in the main text? Why some were annotated, some not? CD14+CD163+DR+ (annotation for the light green cluster) was also the annotation for cluster 3, so why ignoring this cluster? I could not find anywhere information on how and why this decision was made. In addition, in the main text, authors states lines 253-254 that only 5 clusters of monocytes were identified which is misleading and should be rephrased

Our response:

We thank the reviewer for raising this important point and we apologize for not explaining this in sufficient detail. The FlowSOM algorithm in Flowjo requires the user to determine a specific number of metaclusters that should be generated from the data. In the case of the monocyte compartment, 10 clusters were chosen. All clusters that were generated by the algorithm were annotated in Figure 3b and extended data figure 6 with the markers highly expressed in the respective clusters. The two clusters marked with “/” included cells that did not show notable expression of any markers, as visualized in the pie charts represented in the FlowSOM maps and in the heatmap in extended data figure 6g. These clusters were deemed as not relevant for the analysis, as they likely contain non-monocytic cells. Three other clusters were also deemed not relevant for the analysis, as they only included a small number of cells (light green and pink) or cells of an unidentifiable phenotype (orange) as depicted in extended data figure 6h. In the revised version of the manuscript, we now explain these aspects in more detail (line 251-254). We also included additional data to illustrate these aspects (extended data figure 6) and revised the figure legends accordingly.

1b. I have the same questions for B cell clusters shown in Fig S7.

Our response:

We thank the reviewer for raising this point. Similar to the monocyte clustering, for the B cell clusters we expanded the figures in extended data Fig 8 now including the data for all clusters. Here, one cluster (red) was excluded from the analysis, as it mostly contained cells not expressing any of the relevant B cell associated markers considered in this context (Extended Data Fig. 8c). We now revised the current form of the manuscript with additional data as requested and amended the manuscript (line 287-288).

2. The FlowSOM spanning trees shown in Fig 3b, and in Figs S6-S7 do not effectively illustrate marker expression within clusters or differences between groups. For marker expression in clusters, use heatmaps instead, as done in Fig S7a for T cells, or dot plots as in Fig S8a. For cluster composition differences between groups, stick to frequency graphs as shown in Fig S8d, showing individual patient data. Consider removing the tree plots altogether, they take up a lot of space but are difficult to interpret (and not legible at the current resolution).

Our response:

We thank the reviewer for raising this important point. We would like to emphasize that the requested heatmaps, frequency bar graphs and cluster compositions for the FlowSOM spanning trees were already included for all analyses in the revised manuscript submitted in July (new extended data figure 6, f, g, h). In the most recent version of our manuscript, we additionally updated figure 3b by including the pie chart indicating marker expression of the respective clusters. Further, we expanded the extended data sets for monocytes, T and B cell cluster (including all clusters, extended data figure 6, 7, 8), updated figure legends and the manuscript text. For all conventional gating, we now also included (in part requested in point 4) dot plots showing individual data points (extended data figure 6a,d and 7a)

Although we agree that the FlowSOM maps can be slightly overwhelming and difficult to interpret, we believe that they provide valuable insights as they include all relevant information, including marker expression in the clusters and differences between groups, in one figure panel. Due to the nature of the flowSOM method, in which all patient data within the different groups are merged, showing values for individual patients is not feasible^{1,2}. FlowSOM relies on the ability to cluster large data-sets analyzed with several markers in an unbiased fashion, as all cells without any biased gating are included. This enables a clear overview on how cell markers behave on all cells and to detect subsets which might be not detected using conventional evaluation techniques. Further, we believe that the combination of conventional gating (showing individual data points) and cluster analysis (unbiased approach to cluster all cells across a large number of markers), as represented in our manuscript, is a complementary and sophisticated way to analyze multi-color flow cytometry data. For this reason, we would prefer to keep these data in the manuscript and hope that the extensive adjustments made in the most recent version clarify to points raised by reviewer 3.

3. Figure S7a refers to 16 clusters from a tSNE analysis in Fig 3. There is no panel in Fig 3 that is showing a tSNE plot depicting the 16 T cell clusters? In addition, please show the composition differences across groups for all 16 clusters, not only selected clusters 1, 11 and 12.

Our response:

We thank the reviewer for this important point and apologize for not including the information for all 16 clusters. We now extended the extended data with all the information (extended data figure 7b, c and d). As visible in the heatmap in extended data figure 7c, the 16 clusters represent the 4 subtypes (naïve, central memory, effector memory and terminal effector cells) of CD4+ and CD8+ T cells, with high or low expression of the activation markers PD-1 and TIGIT, respectively. To simplify the visualization of these clusters, the TIGIT/PD-1 high and low clusters were merged for the tSNE representation in Figure 3i, resulting in 8 different displayed clusters. The differences of TIGIT/PD-1 expression is also visible in Figure 3i. Further, we now inserted the complete information for all 16 clusters, including t-SNE, heatmap and frequency bar graph (extended data figure 7b, c, d).

4. The cluster composition graphs in Fig 3a, Fig S6bf and S7cfj should show individual datapoints for each patient.

Our response

We thank the reviewer for raising this point and according to the reviewer's suggestion we included graphs showing individual data points. The new extended data 6a shows the individual data points for figure 3a (conventional gating). Extended data 6d shows the individual data points for extended figure 6c (old extended data figure 6b, conventional gating). In addition to the reviewer's suggestions, we also included the individual data points for figure 3g, which is now depicted in extended data figure 7a. The source data file has been updated accordingly.

For all cluster analysis (referring also to point 2): Due to the nature of the flowSOM method, where all patient data within the different groups are merged, showing values for individual patients is not feasible^{1,2}. FlowSOM relies in the ability to cluster large data-sets analyzed with several markers in an unbiased fashion, as all cells without any biased gating are included. This enables a clear overview on how cell markers behave on all cells and to detect subsets which might be not detected as such. Further, we believe that the combination of conventional gating (showing individual data points) and cluster analysis (unbiased approach to cluster all cells across a large number of markers), as represented in our manuscript, is a complementary and sophisticated way to analyze multi-color flow cytometry data.

Other concerns

5. Biological sex information is not available in Table S1 as indicated in methods lines 609-610.

Our response:

We thank the reviewer for raising this point. Table S1 has been now updated accordingly.

6. Correlation analysis lines 314-316: indicate which datasets were compared (i.e., single-cell RNAseq compared to flow cytometry and bulk RNAseq)

Our responses:

We thank the reviewer for this comment. We compared all three data-sets with each other and observed a very good correlation. We updated the revised version accordingly to clarify this point (line 315-318).

7. It is not clear why a PTB cohort was not included for comparison in Fig 5i/ Fig S12. It should be at least mentioned in the discussion that whether the aCD4m T cell subset is EPTB specific, or general TB-specific, remain to be determined.

Our response:

We thank the reviewer for this important comment. According to the reviewer's suggestion we included a sentence within the discussion, clarifying that the TB-specificity of aCD4Tm needs to be determined in the future (line 504) ³.

8. Lines 533-534: this is a stretch given the current study design. The standard workflow for identifying diagnostic signatures is to have a much larger cohort, and divide it into training, test and validation sets. This should be the next step. Please indicate that further validation of the diagnostic signature is warranted before moving into clinical trials. Include in the limitations of the study as well.

Our responses:

We thank the reviewer for raising this important point and agree that further validations are needed before moving to larger scale clinical trials. Therefore, as suggested by the reviewer, we mention this limitation of the study in the respective section of the discussion (line 536-539). To be more specific, in 2026 we will evaluate data of a larger, multicenter study with a similar design which will also provide longitudinal data. The data of the current study can be exploited as training set for this study but also for other groups working on similar cohorts.

9. The figures are at too low resolution and the font is too small. It is not always possible to decipher the + from the -, and some of the writing.

Our response:

We thank the reviewer for this comment and adjusted the font and resolution throughout the manuscript.

10. Line 271, replace lymphocyte with T cell? NK cells (discussed in the previous paragraph) are also lymphocytes.

Our response:

We thank reviewer three for highlighting this. This has been now changed.

11. Typo line 333: should be Fig 10d?

Our response:

This has now been changed.

12. Lines 513-520, missing reference.

Our response:

The reference has now been added.

Reviewer #3 (Remarks on code availability):

The link for the code is invalid.

Our response:

We apologize for the inconvenience related to the repository link. Since we are only permitted to make the GitLab repository (<https://gitlab.dzne.de/dahmk/EPTB-RNAseq-Analysis>) publicly available upon publication, we have temporarily deposited the repository content on Sciebo, an academic data cloud. The data can be accessed using the following link and password:

<https://uni-bonn.sciebo.de/s/cG6GPrHgeZ8Nk7V> (Password: 7HSS3sG4NJ)

References

- 1 Quintelier, K. *et al.* Analyzing high-dimensional cytometry data using FlowSOM. *Nat Protoc* **16**, 3775-3801 (2021). <https://doi.org:10.1038/s41596-021-00550-0>
- 2 Van Gassen, S. *et al.* FlowSOM: Using self-organizing maps for visualization and interpretation of cytometry data. *Cytom Part A* **87a**, 636-645 (2015). <https://doi.org:10.1002/cyto.a.22625>
- 3 Pan, J. H. *et al.* Research progress of single-cell sequencing in tuberculosis. *Frontiers in Immunology* **14** (2023). <https://doi.org:10.3389/fimmu.2023.1276194>